# `MetaSeal`: Defending Against Image Attribution Forgery Through Content-Dependent Cryptographic Watermarks

**Tong Zhou**                                                                 *zhou.tong1@northeastern.edu*
*Northeastern University*

**Ruyi Ding**                                                                  *ding.ruy@northeastern.edu*
*Northeastern University*

**Gaowen Liu**                                                                 *gaoliu@cisco.com*
*Cisco*

**Charles Fleming**                                                            *chflemin@cisco.com*
*Cisco*

**Ramana Rao Kompella**                                                        *rkompell@cisco.com*
*Cisco*

**Yunsi Fei**                                                                  *y.fei@northeastern.edu*
*Northeastern University*

**Xiaolin Xu**                                                                 *x.xu@northeastern.edu*
*Northeastern University*

**Shaolei Ren**                                                               *shaolei@ucr.edu*
*University of California, Riverside*

**Reviewed on OpenReview:** *https://openreview.net/forum?id=8i3ErmCfdJ*

## Abstract

The rapid growth of digital and AI-generated images has amplified the need for secure and verifiable methods of image attribution. While digital watermarking offers more robust protection than metadata-based approaches–which can be easily stripped–current watermarking techniques remain vulnerable to forgery, creating risks of misattribution that can damage the reputations of AI model developers and the rights of digital artists. The vulnerabilities of digital watermarking arise from two key issues: (1) content-agnostic watermarks, which, once learned or leaked, can be transferred across images to fake attribution, and (2) reliance on detector-based verification, which is unreliable since detectors can be tricked. We present `MetaSeal`, a novel framework for content-dependent watermarking with cryptographic security guarantees to safeguard image attribution. Our design provides (1) **forgery resistance**, preventing unauthorized replication and enforcing cryptographic verification; (2) **robust self-contained protection**, embedding attribution directly into images while maintaining robustness against benign transformations; and (3) **evidence of tampering**, making malicious alterations visually detectable. Experiments demonstrate that `MetaSeal` effectively mitigates forgery attempts and applies to both natural and AI-generated images, establishing a new standard for secure image attribution. Code is available at: `https://github.com/Tongzhou0101/MetaSeal`.

# 1 Introduction

As digital content creation and sharing accelerate, especially with the rise of AI-generated content (AIGC), securing the attribution of visual content has become essential for the entire digital ecosystem (Wang et al., 2024; Zhao et al., 2025; Knott et al., 2024). For content creators, the lack of reliable attribution methods opens the door for bad actors to fake their creations, causing financial loss (Korus, 2017; Lindley, 2020). For AI model developers, misattribution of AI-generated content can lead to reputation damage, as they are often held accountable for the outputs of their models (Jovanović et al.; Zhou et al., 2024). Notably, emerging regulations such as the EU AI Act explicitly assign accountability to AI developers for harmful or misleading content generated by their models (ISACA, 2024), raising the stakes for accurate attribution.

In response, metadata-based methods and watermarking techniques are widely used to identify the source of images. However, metadata-based methods, such as the C2PA standard (Rosenthol, 2022), are fragile; metadata can be stripped or corrupted through common processes like reformatting or transmission, leaving content without any traceable attribution (Korus, 2017). Digital watermarking, in contrast, offers more robust protection by embedding information directly into images (Zhu et al., 2018; Fernandez et al., 2023; Zhang et al., 2024).

However, existing watermarking methods fall short of supporting reliable image attribution. Current techniques are mostly designed for two distinct purposes: copyright protection and image authentication. Copyright-oriented watermarking focuses on robustness, aiming to ensure that the watermark survives adversarial removal attempts. These techniques have been adapted for detecting AIGC by embedding predefined watermarks either through post-processing (Xu et al., 2025) or directly during content generation (in-processing) (Fernandez et al., 2023; Wen et al., 2023). In contrast, authentication watermarking emphasizes fragility, aiming to detect any modification made to an image (Lu & Liao, 2001; Zhang et al., 2024; Sander et al., 2025). *Attribution, however, introduces a fundamentally different requirement: the system must prevent images from being falsely linked to incorrect sources.* Unfortunately, recent studies have shown that even state-of-the-art methods remain highly vulnerable to forgery attacks (Saberi et al., 2024; Yang et al., 2024a), which undermines their ability to ensure trustworthy attribution, as illustrated in Fig. 1.

This vulnerability stems from two key factors: (1) the use of *content-agnostic watermarks* and (2) reliance on *detector-based verification*. Content-agnostic watermarks, such as fixed patterns, apply the same watermark to all images, regardless of their unique content (Bui et al., 2023; Wen et al., 2023). This approach leaves watermarks susceptible to forgery, as these image-independent patterns can be extracted and replicated across unrelated images, falsely implying attribution (Yang et al., 2024a). Besides, the reliance on detector-based source identification is inherently weak since these detectors are vulnerable to adversarial attacks (Saberi et al., 2024). Attackers can craft small, imperceptible

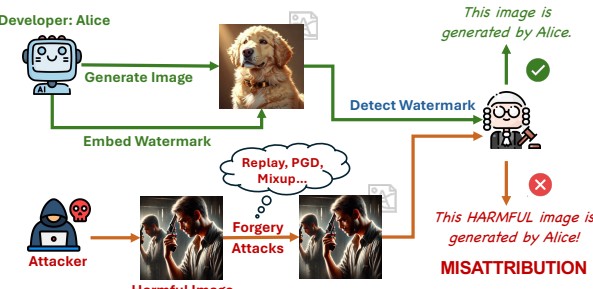

Figure 1: Attackers can forge watermarked images that falsely attribute harmful or manipulated content to a model, risking developer reputation.

modifications that can cause the detector to falsely recognize an irrelevant image as an authentic watermarked one. Such forgery attacks not only weaken security guarantees but also erode trust in attribution outcomes.

These failures highlight two critical open questions for designing watermarking schemes for attribution:

- **What to embed** to ensure the watermark is bound securely to its rightful source?
- **How to embed and verify** to prevent forgery?

Addressing these questions is essential to advance watermarking schemes from general-purpose protection mechanisms toward robust and accountable attribution systems.

**This work**: We propose `MetaSeal`, an image attribution watermark that provides innovative solutions to the aforementioned fundamental questions under a forgery-centric threat model. Our key insight is that, to escape the forgery trap, the reliable attribution requires both **content-dependent watermarking** and **cryptographic verification guarantees**, with the attribution information being self-contained and provable. Specifically, we generate watermarks tailored to image contents using digital signatures and embed them directly into the image. By employing cryptographic verification instead of conventional detectors, `MetaSeal` enhances security and effectively mitigates the risk of forgery attacks.

Embedding content-dependent watermarks that support cryptographic verification poses significant technical challenges. A critical requirement is achieving perfect signature extraction accuracy—an area where current embedding techniques often fall short, particularly when the embedding capacity is large (Fernandez et al., 2023; Zhu et al., 2018), as demonstrated in Sec. 3.2. Moreover, cryptographic verification depends on the integrity of the digital signature (Schneider & Chang, 1996). However, images are often subjected to various transformations, which can compromise the embedded signature's validity. For real-world applicability, the watermark must remain resilient to benign transformations, such as JPEG compression, to ensure its effectiveness and practicality.

To address these challenges, `MetaSeal` provides a principled integration of semantic binding with cryptographic verification and exact extraction. First, it generates content-dependent signatures that encapsulate the image's semantic information, ensuring the watermark is tied to the content rather than pixel details. This design ensures the watermark remains consistent under benign transformations that preserve the image's semantics while being invalidated by malicious perturbations revealed by tampering evidence. Second, to enhance extraction performance, `MetaSeal` transforms cryptographic signatures into visually meaningful patterns (e.g., QR codes) and employs an invertible embedding and extraction process. This approach improves the robustness and accuracy of signature recovery, ensuring the watermark remains intact and functional even under challenging conditions. Our contributions are summarized as follows:

1. We introduce `MetaSeal`, an image attribution watermark that explicitly addresses the foundational questions of **what to embed** and **how to embed/verify** under a **forgery-centric threat model**, by binding content-dependent cryptographic signatures directly into images.

2. We present a **publicly verifiable** attribution framework through the proposed *Visual Attribution Signature*. By integrating asymmetric cryptography into watermarking, `MetaSeal` enables any third party to verify attribution using a public key, without relying on secret detectors or private verification.

3. `MetaSeal` maintains robustness against benign (e.g., compression) transformations to a certain extent while providing tampering evidence of malicious perturbations through visual artifacts.

4. `MetaSeal` achieves perfect extraction accuracy with payloads $88\times$ larger than baseline methods, maintaining promising image quality and supporting both natural and AI-generated content, advancing reliable image attribution.

## 2 Background

### 2.1 Image Attribution

Securing image attribution has grown increasingly critical with the rise of AIGC. Two primary approaches, i.e., metadata-based methods and watermarking, offer varying levels of effectiveness in addressing this challenge. Metadata-based methods, such as those standardized by C2PA (Rosenthol, 2022), embed verification information directly into image files. This typically involves creating a digital signature using a private key to sign a hash of the content, with the corresponding public key used for verification by comparing the decrypted hash to a newly computed one (Tonkin & Allinson, 2006). Although effective in preserving integrity under ideal conditions, these methods are highly fragile. Metadata can be easily lost during format conversions or minor edits, rendering the attribution unverifiable. In contrast, watermarking techniques embed verification information into the image's pixel (Begum & Uddin, 2020), providing greater resilience across transformations, and becoming a more robust solution for content attribution.

Besides, some methods have been proposed superficially for AIGC attribution, which train classifiers that exploit distributional differences across generative models (Yu et al., 2019; Girish et al., 2021; Sha et al., 2023). While effective for distinguishing between models, these approaches offer no verifiable proof and are limited to AIGC attribution. In contrast, our approach offers a general-purpose image attribution framework that is not limited to AIGC and provides strong, cryptographically verifiable evidence of provenance, making it applicable across a broad range of use cases.

## 2.2 Image Watermarking

Watermarking techniques differ significantly in design goals, depending on whether they are used for copyright protection, image authentication, or image attribution. Table 1 summarizes the differences across these goals, highlighting distinctions in detection methodology, robustness requirements, and security risks. More related works and discussions on image watermarking are provided in Sec. 8.

**Copyright protection** Watermarking for copyright protection primarily aims to assert ownership and resist removal. Traditional methods based on signal processing techniques (e.g., DWT-DCT) embed watermarks in frequency domains to achieve robustness against common manipulations (Barni et al., 2001; Cox et al., 2007). More recent deep learning-based approaches, such as HiDDeN (Zhu et al., 2018) and RivaGAN (Zhang et al., 2019), use encoder-decoder architectures and adversarial training to embed fixed-length bitstreams (typically under 100 bits) that survive a wide range of distortions.

Besides, some image watermarking methods have been proposed specifically for detecting AIGC—that is, embedding watermarks into generated images to identify whether they were produced by a particular model (Wen et al., 2023; Fernandez et al., 2023; Yang et al., 2024b). By emphasizing robustness against removal, such methods resemble copyright protection watermarks, implicitly treating the model owner as the copyright holder. Typically, they embed fixed, model-specific watermarks through modifying model weights (Fernandez et al., 2023; Kim et al., 2024) or the generation process (Wen et al., 2023; Yang et al., 2024b). However, their content-agnostic design introduces a critical security risk: attackers can carry out forgery attacks by extracting and transplanting the watermark onto unrelated images, leading to misattribution (Yang et al., 2024a; Saberi et al., 2024). By overlooking this risk, such methods cannot be considered reliable attribution watermarks. This issue is especially concerning because model owners do not legally hold the copyright to generated content, yet they may still be held accountable for malicious generations.

**Authentication** Authentication watermarking schemes are intentionally designed to be fragile, breaking when the watermarked content undergoes unauthorized modifications. This fragility serves as an integrity check mechanism. Verification typically requires access to embedding secrets or reference watermarks for comparison. For instance, classical authentication watermarks are often embedded in specific wavelet coefficient locations, where the location secrets will be revealed during the authentication process (Lu & Liao, 2001). Beyond basic authentication, advanced methods for tampering localization have been developed (Hur-

Table 1: Comparison of watermarking goals: Copyright Protection, Authentication, and Attribution.

| Aspect | Copyright Protection | Authentication | Attribution (Ours) |
|---|---|---|---|
| **Objective** | Assert ownership or detect AIGC | Detect Modification | Trace rightful creator or source |
| **Detection Method** | Compare with pre-defined watermarks with error tolerance | Check integrity via exact match with ground truth | Extract proof then apply cryptographic verification |
| **Detection Entity** | Content owner | Content Recipient | Any party with public verification keys |
| **Robustness** | High robustness against removal attacks | High sensitivity to editing | Robust against benign transformation but not semantic tampering |
| **Security Risk** | Attackers aim to remove watermark to erase ownership | Attackers modify content while evading detection | Attackers forge watermark to misattribute ownership |

rah et al., 2019; Kamili et al., 2020; Zhang et al., 2024; Sander et al., 2025) that not only detect modifications but also precisely identify which regions of an image have been altered. These techniques provide detailed information about the modifications when authenticity is compromised, offering a more comprehensive integrity assessment than simple binary authentication.

**Attribution** Unlike copyright protection watermarks, which aim to *assert* ownership, attribution watermarks are designed to *prove* the rightful creator or source of an image, addressing a distinct and critical security risk: forgery attacks. There exists an inherent trade-off between robustness and unforgeability: increasing robustness to transformations/removal attacks expands the space of modified images that successfully verify, which can inadvertently increase the risk of misattribution. Thus, attribution watermarking requires a different balance than copyright protection—**prioritizing forge-resistance while maintaining sufficient robustness for legitimate use cases**, rather than maximizing robustness against all possible removal attacks. This watermark can be used to regulate the image generation models to tell if a malicious image is really generated from it. To improve trustworthiness, it should offer public verifiability.

Besides, for image attribution, removing the watermark merely renders the attribution mechanism ineffective, preventing identification of the image's source. This limitation poses minimal harm to image generation service providers or digital artists, especially since removal attempts typically compromise image quality, making the altered content less valuable or usable. A more significant threat lies in forgery attacks that deceive watermark verifiers into classifying unauthorized images as authentically generated by a specific source. Such attacks could lead to service providers being falsely accused of inadequate safety mechanisms or enable bad actors to counterfeit an artist's work, potentially causing financial harm. *Thus, forgery attacks pose more severe consequences than removal attacks in this context, highlighting the need to strengthen watermarking mechanisms against forgery.*

### 2.3 Content-dependent Techniques

One of the key vulnerabilities in prior watermarking schemes is their content-agnostic nature, where the watermark embedded in one image can be extracted and transplanted into another to create falsely authenticated content. One mitigation is to make the watermark content-dependent, creating an intrinsic link between the watermark and the specific image content.

Content-dependency can be achieved using various hashing techniques, including cryptographic hashes like MD5/SHA-256 (Sobti & Geetha, 2012), or perceptual image hashes such as NeuralHash (Farid, 2021). Cryptographic hashes require exact bit-by-bit matches to validate, making them suitable for strict authentication but overly sensitive to benign modifications. In contrast, perceptual image hashes are designed to produce similar hash values for visually similar images. However, perceptual hashing remains vulnerable to adversarial manipulation, as researchers have demonstrated methods to create hash collisions between visually dissimilar images (Struppek et al., 2022).

In addition, these hashing approaches normally require external storage and are not self-contained within the image itself. Even when embedded as watermarks, they primarily serve for content authentication rather than attribution (Roy et al., 2023; Hussan et al., 2022). A hash can verify that content has not been altered, but cannot independently establish who created it or which system generated it. Overall, these techniques cannot be directly applied to image watermarking to achieve reliable attribution.

## 3 Preliminaries

### 3.1 Watermark Forgery

We categorize common watermark forgery attacks by the modality of their strategies: whether they exploit the embedding process (replay attacks), estimate and reuse the watermark signal (mixup attacks), or directly attack the detection mechanism (PGD attacks).

**Replay Attacks.** Attackers in this case can be dishonest watermark verifiers who know everything required for detection, including the embedding algorithm $\mathcal{E}(I, w)$, the detection algorithm $\mathcal{D}(I)$, the secret key (if

any), and the embedding locations. Given a legitimate watermarked image $I_w = \mathcal{E}(I, w)$, the attacker extracts the watermark $w$ and re-embeds it into a different image $I'$ to forge $I'_w = \mathcal{E}(I', w)$, such that $\mathcal{D}(I'_w) = \text{True}$. Content-agnostic schemes are especially vulnerable, like DCT-based watermarking (Cox et al., 2007), as the static watermark $w$ is transferable across images.

**Mixup Attacks.** Attackers first estimate the watermark signal by computing the average residual between $n$ watermarked images $I_w$ and their original versions $I$:

$$\hat{w} = \frac{1}{n} \sum_{i}^{n} (I_w^i - I^i) \tag{1}$$

The extracted signal $\hat{w}$ is then added to a new image $I'$ to forge a watermarked version:

$$I'_w = I' + \hat{w} \tag{2}$$

Such attacks are effective when the watermark is additive and not tightly bound to the original content (Xu et al., 2025).

For generative watermarking schemes like Tree-Ring (Wen et al., 2023), the attacker can prompt the generative model to synthesize a white noise image $I^{\text{noise}}$ (Saberi et al., 2024). The corresponding watermarked output $I_w^{\text{noise}}$ is then linearly blended with a clean image $I'$:

$$\tilde{I} = \lambda I_w^{\text{noise}} + (1 - \lambda)I' \tag{3}$$

where $\lambda \in [0, 1]$ controls the mixing ratio. The forged image $\tilde{I}$ may pass the watermark detector with $\mathcal{D}(\tilde{I}) = \text{True}$.

**PGD Attacks.** These attacks exploit the vulnerability of watermark detectors that are implemented as deep neural networks. Specifically, an attacker aims to manipulate the input image in a minimally perceptible way such that the detector outputs an incorrect prediction. Given a clean, unwatermarked image $I$, the attacker uses the Projected Gradient Descent (PGD) algorithm (Madry et al., 2018) to construct an adversarial example $I_{\text{adv}} = I + \delta$, where the perturbation $\delta$ is carefully optimized to induce a misclassification by the detector. Formally, the attacker solves the following optimization problem:

$$\min_{\delta} \mathcal{L}\left(\mathcal{D}(I + \delta), y\right) \quad \text{subject to} \quad \|\delta\|_\infty \le \epsilon \tag{4}$$

where $\mathcal{L}$ is the loss function (typically cross-entropy), $y$ is the target label desired by the attacker (e.g., $y = 1$ to indicate a watermarked image in binary classification, or $y = w$ when using ground-truth watermark bit strings), and $\epsilon$ bounds the perturbation magnitude (e.g., in the $\ell_\infty$ norm) to ensure imperceptibility.

Such adversarial perturbations are often visually indistinguishable from the original image but can reliably mislead the detector into producing incorrect results, such as detecting a watermark where none exists. These attacks have been demonstrated to be effective under both white-box and black-box threat models (Saberi et al., 2024; Zhao et al., 2025), highlighting the need for trustworthiness in neural watermark detectors.

### 3.2 Limitation of Prior Works

One of the key aspects to defend against forgery attacks via watermarking is to incorporate a cryptographic signature, which in turn requires the watermarking method to support a large embedding capacity. However, we found that current learning-based methods struggle to meet this requirement.

Here, we use HiDDeN (Zhu et al., 2018), a representative learning-based image watermarking framework, as a concrete example to demonstrate this limitation. HiDDeN consists of an end-to-end trainable pipeline comprising an encoder, decoder, and a noise layer. It is trained to minimize message recovery error, maximize image fidelity (i.e., ensure that the image quality does not degrade significantly due to watermark embedding), and enhance robustness against various transformations. Given a cover image $I \in \mathbb{R}^{H \times W \times C}$ and a binary message $m \in \{0, 1\}^k$, the encoder embeds $m$ into $I$ to generate a watermarked image $I_w$. The decoder attempts to recover the message from a possibly distorted version of $I_w$. Detection is typically performed

by decoding a message $\hat{m}$ from the watermarked image and comparing it to the expected message $m$; the image is considered watermarked if the *bit error rate (BER)* is below a certain threshold $\tau$:

$$\text{BER}(m, \hat{m}) = \frac{1}{k} \sum_{i=1}^{k} \mathbf{1}[\hat{m}_i \neq m_i] \leq \tau, \quad \hat{m}_i = \mathbf{1}(p(\hat{m}_i) \geq 0.5). \tag{5}$$

where $p(\hat{m}_i)$ is the predicted logit of $\hat{m}_i$. It originally only supports a small payload, e.g., a message length of 30 bits.

Considering that a cryptographic signature is typically larger than 512 bits, we test whether HiDDeN can be adapted to support such a large payload, enabling integration of cryptographic signatures to enhance security against forgery. However, as illustrated in Fig. 2, HiDDeN fails to converge on accurate message recovery when trained with a message of 512 bits. While image fidelity continues to improve during training, the BER remains high, indicating a fundamental limitation in embedding capacity. Furthermore, existing approaches such as Stable Signature (Fernandez et al., 2023) and HiDDeN only support the embedding and extraction of

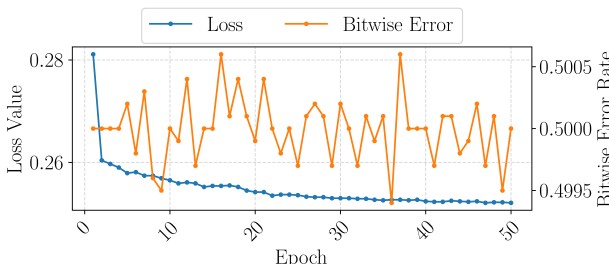

Figure 2: Training performance of HiDDeN with 512-bit messages: loss optimizes image fidelity while recovery accuracy remains low (unconverged BER).

a fixed watermark (i.e., a fixed set of message bits) once trained. If the message changes, the model requires retraining or fine-tuning. Since cryptographic signatures are content-dependent and unique to each instance, this approach is impractical for signature integration, as it is infeasible to retrain the model for every new signature.

## 4 Problem Formulation

### 4.1 Threat Model

**Attackers.** We consider attackers whose objective is to forge watermarks onto unrelated images in order to falsely claim authorship or misrepresent the provenance of content. Such attackers may aim to undermine the credibility of creators or model developers by linking their names to malicious or low-quality works, or fake some creators' work with valid watermarks for financial gain. We assume the attacker has full knowledge of the watermarking algorithm, including the embedding and detection procedures, and access to a set of watermarked samples. However, the adversary does not possess internal parameters of the encoder or decoder models (i.e., operates under a gray-box threat model). This is a reasonable assumption since watermarking algorithms are typically open-sourced, while encoder/decoder models are commonly deployed as black-box services on cloud platforms for practical usage.

**Protectors.** Protectors focus on ensuring reliable attribution by securely linking content to its rightful creator or generator through robust watermarks. The watermark must withstand benign transformations, maintaining its integrity under non-adversarial conditions. Crucially, it must also provide forgery resistance, preventing adversaries from successfully embedding convincing but unauthorized watermarks into unrelated content.

### 4.2 Design Requirements

We outline the primary requirements for building a reliable watermarking framework for image attribution:

- **Content Dependency:** The watermark must be tied to the content of each image to prevent easy estimation and reuse across unrelated images. This ensures that attribution remains tightly coupled to the image's inherent characteristics.

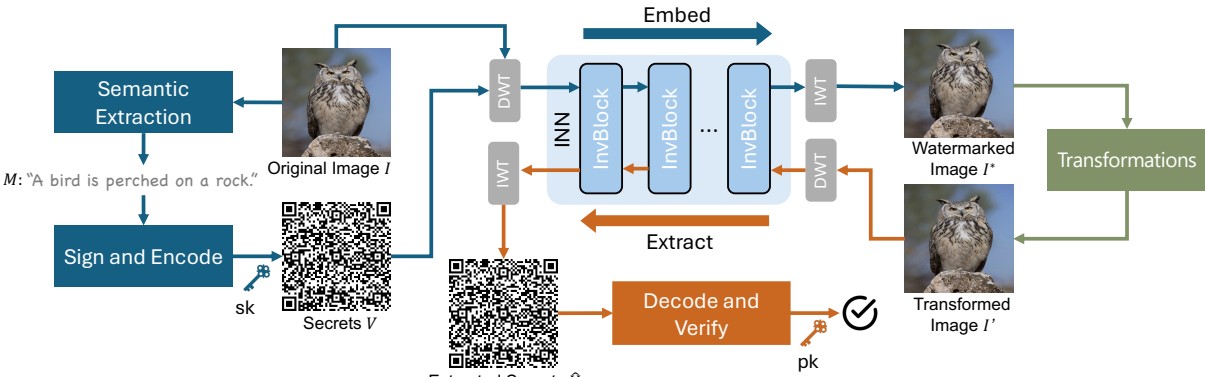

Figure 3: The inference process of `MetaSeal`. **Embedding:** Semantic features are extracted to generate a cryptographic signature using the private key sk, which is encoded into a visual pattern and embedded into the image using an invertible neural network (INN, trained with Eq. 15). The resulting watermarked image may undergo transformations such as JPEG compression. **Extraction:** The embedded secret is recovered using the same INN and then decoded to verify attribution using the public key pk. The secret is embedded in the frequency domain using discrete wavelet transform (DWT) and Inverse Wavelet Transform (IWT) for improved robustness and imperceptibility.

- **Cryptographic Verification:** The watermark should provide cryptographic guarantees, allowing mathematically secure verification to protect against forgery attacks. This approach eliminates reliance on vulnerable detector-based systems and ensures attribution integrity.

- **Self-Contained Attribution:** All necessary attribution information should be embedded directly into the image, avoiding external dependencies such as metadata that can be stripped or corrupted. This ensures the attribution remains intact even after common image transmissions.

### 4.3 Challenges

To meet these requirements, cryptographic signatures offer a promising foundation since the signature is dependent on the content and provides a cryptographic security guarantee. However, two key challenges must be addressed:

**Challenge 1: Content Dependency vs. Robustness.** *How can the watermark be made content-dependent to prevent forgery, yet robust against benign transformations?* Watermarks that are tightly coupled to image content reduce the risk of being copied across unrelated images. However, if the dependency is too strict—such as signing pixel-level details—the watermark may become fragile, breaking under benign transformations. A practical solution must balance these factors: ensuring the watermark remains valid under standard modifications, but invalid under adversarial changes that alter the image's meaning.

**Challenge 2: Payload Size vs. Extraction Accuracy.** *How can the cryptographic signature's large payload be precisely extracted while balancing embedding capacity and extraction accuracy?* Cryptographic signatures often involve a significant amount of data, which challenges existing embedding techniques to achieve both high capacity and precise extraction. The trade-off between embedding capacity and extraction accuracy is critical, as inaccuracies in extraction could compromise the validity of cryptographic verification.

## 5 Image Attribution Watermark: `MetaSeal`

To safeguard image attribution via watermarking, we propose `MetaSeal`, which addresses two fundamental questions illustrated in Fig. 3. First, regarding **what to embed**, we advocate for the use of cryptographic signatures to create content-dependent watermarks, moving away from fixed patterns. These signatures ensure the watermark is tied to the image content, mitigating forgery attacks. Second, regarding **how to embed/verify**, we emphasize the critical need for exact secret extraction. To achieve this, we transform

cryptographic signatures into meaningful visual patterns, enhancing robustness against benign transformations. For verification, instead of relying on binary detectors (Saberi et al., 2024) or statistical tests (Wen et al., 2023), we adopt an invertible process of the embedding mechanism to retrieve the embedded cryptographic signature from the watermarked image, promoting precise verification.

The remainder of this section is organized as follows: We overview the proposed scheme in Sec. 5.1, then we present how our solutions overcome the identified challenges in Sec. 5.2 and Sec. 5.3 to complete our scheme.

### 5.1 Visual Attribution Signature

We propose a cryptographically grounded attribution scheme that integrates digital signature algorithms into image watermarking via structured visual patterns. Unlike metadata, which is fragile and easily stripped, our method embeds signatures directly into images as spatially redundant visual structures (e.g., QR codes). This self-trained design improves the effectiveness of watermark verification.

**Definition 1** (Visual Attribution Signature). *Let $I \in \mathbb{R}^{H \times W \times 3}$ denote an input image and $M = f(I)$ be the semantic features extracted by a function $f$. The visual attribution signature consists of the following components:*

- ***KeyGen***$(1^\lambda) \to (\mathsf{sk}, \mathsf{pk})$*: Generates a public-private key pair under a cryptographic scheme, where $\lambda$ is the security parameter.*

- ***Sign***$(\mathsf{sk}, M) \to S$*: Computes the digital signature $S$ for $M$ using the private key $\mathsf{sk}$.*

- ***PatternEnc***$(M, S) \to V$*: Encodes $M$ and $S$ into a structured binary pattern $V \in \{0, 1\}^{H \times W}$.*

- ***Embed***$(I, V) \to I^\star$*: Embeds $V$ into $I$ to produce a watermarked image $I^\star$.*

- ***Extract***$(I') \to \hat{V}$*: Extracts an estimated pattern $\hat{V}$ from a potentially modified image $I'$.*

- ***PatternDec***$(\hat{V}) \to (\hat{M}, \hat{S})$*: Decodes $\hat{V}$ to get recovered message-signature pair $(\hat{M}, \hat{S})$.*

- ***Verify***$(\mathsf{pk}, \hat{M}, \hat{S}) \to \{\mathsf{true}, \mathsf{false}\}$*: Accepts if $\hat{S}$ is a valid signature of $\hat{M}$ under the public key $\mathsf{pk}$.*

Crucially, unlike conventional digital signatures that assume direct access to the signed message $M$ and verify $S$ by checking if $\mathsf{pk}(S) = M$, watermarking complicates this process. Since embedding alters the image, the extracted features may not match the original, i.e., $f(I^\star) \neq f(I) = M$ is not guaranteed (Korus, 2017; Fairoze et al.), making $M$ inaccessible from $I^\star$ alone via $f$. To ensure correct verification, our scheme requires exact recovery of both $M$ and $S$, i.e., $\hat{M} = M$ and $\hat{S} = S$, such that $\mathsf{pk}(\hat{S}) = \hat{M}$. To achieve this, we jointly encode $(M, S)$ into a visual pattern $V$ via **PatternEnc**, shifting the verification requirement to accurate visual pattern extraction, while $f(I') \approx M$ can be used as supplementary verification.

In our work, we use QR code as the visual pattern $V$ since it can store a significant amount of information in a small, visually compact area. This makes them ideal for embedding large payloads. Also, it includes built-in error correction mechanisms that allow for data recovery even if part of the code is damaged or obscured, and it supports easy and reliable scanning by machines like smartphones.

This security guarantee of the proposed visual attribution signature against forgery attacks follows directly from the existential unforgeability of the underlying digital signature scheme (e.g., ECDSA) (Goldwasser et al., 1988). We provide further analysis and demonstrations of resistance to adaptive forgery attacks in Sec. 6.5.

Beyond cryptographic soundness, our structured visual pattern design offers significant advantages over traditional bit-wise embedding approaches. First, it achieves superior error resilience through both local spatial correlation and global error correction codes, enabling robust signature recovery. Second, the structured pattern provides built-in redundancy to support error correction. The detailed analysis is presented in Appendix A. Furthermore, the decoded pattern could provide visual feedback indicating tampering attempts (see Sec. 6.4).

## 5.2 Semantic Extraction

To address *Challenge 1*, the ideal feature extractor $f$ should satisfy two complementary properties:

a) **Content Dependency:** The extracted features $M = f(I)$ should uniquely characterize each image's semantic content, ensuring that images with genuinely different semantics yield different features:

$$\text{if } \text{Sem}(I_1) \neq \text{Sem}(I_2) \implies f(I_1) \neq f(I_2), \tag{6}$$

where $\text{Sem}(\cdot)$ denotes the semantic content of the image.

b) **Transformation Robustness:** The features should remain invariant under benign transformations $\mathcal{T}(\cdot)$ (e.g., compression, resizing, or color shifts) while changing under adversarial manipulations $\mathcal{A}(\cdot)$ that alter semantics:

$$f(\mathcal{T}(I)) = f(I), \quad f(\mathcal{A}(I)) \neq f(I). \tag{7}$$

By distinguishing *semantic differences* from *superficial transformations*, $f$ balances robustness and security: if $M$ is too coarse, it risks reuse across unrelated images; if it is too tied to pixel-level details, it breaks under benign edits. By focusing on semantics, our method distinguishes malicious manipulations from benign changes while minimizing the risk of watermark reuse (see Sec. 6.5).

To realize this, we implement $f$ using an image-to-text model that maps visual content to a deterministic textual description, which serves as the high-level semantic summary $M$ and is more invariant to benign image transformations than pixel-level features. Specifically, we decompose $f$ into an encoder–decoder architecture:

$$M = f_{\text{dec}}(f_{\text{enc}}(I)), \tag{8}$$

where $f_{\text{enc}}$ is a vision model that extracts high-level visual features, and $f_{\text{dec}}$ is a language model that generates a textual description. This architecture ensures that $M$ captures semantic content while being robust to pixel-level variations. In practice, $f$ can be instantiated as a pre-trained image captioning model. When integrated with our visual signature scheme, this semantic representation provides a robust basis for attribution: it mitigates forgery by enforcing content dependency while remaining effective under common benign transformations.

## 5.3 Invertible Embedding and Extraction

To address *Challenge 2*, **Embed** must support a large payload to seamlessly integrate $V$ into the original image $I$. Additionally, **Extract** is expected to be the inverse of **Embed**, ensuring accurate retrieval of the secret $V$. To achieve these properties, we identify invertible neural networks (INNs) as a promising solution.

INNs rely on bijective transformations to ensure perfect input reconstruction through invertible blocks (InvBlocks), a distinct advantage over traditional neural networks where operations are rarely reversible (Jing et al., 2021; Xing et al., 2021; Xiao et al., 2020). While this reversibility has been exploited to achieve high-capacity steganography (Xing et al., 2021; Lu et al., 2021) and imperceptible watermarking (Ma et al., 2022), we leverage it here for a different purpose: enabling exact recovery of large, structured attribution signatures required for public cryptographic verification, while inherently providing editing localization as tampering evidence.

Within each invertible block $i$, the inputs include the image component $I^i$ and the secret $V^i$. The forward (embedding) operation of block $i$ is defined as:

$$I^{i+1} = I^i + \phi(V^i), \tag{9}$$
$$V^{i+1} = V^i \odot \exp\left(\alpha(\rho(I^{i+1}))\right) + \eta(I^{i+1}), \tag{10}$$

where $\phi(\cdot)$, $\rho(\cdot)$, and $\eta(\cdot)$ are learnable modules (e.g., convolutional or dense layers), $\odot$ denotes element-wise multiplication, and $\alpha$ is a sigmoid-based clamping function to stabilize scaling. The design ensures that both the image and the secret can be perfectly recovered using the inverse operation:

$$V^i = \left(V^{i+1} - \eta(I^{i+1})\right) \odot \exp\left(-\alpha(\rho(I^{i+1}))\right), \tag{11}$$
$$I^i = I^{i+1} - \phi(V^i). \tag{12}$$

To better understand how it works in our framework, let $g_\theta$ denote the INN parameterized by $\theta$. The forward embedding process using the INN is simplified as:

$$(I^\star, r) = g_\theta(I, V), \tag{13}$$

where $I^\star$ represents the watermarked image, and $r$ denotes the residual information that cannot be seamlessly embedded into $I$ due to the high hiding capacity. The residual $r$ is always modeled as an image-agnostic Gaussian distribution, ensuring that $I^\star$ alone suffices for accurate recovery (Jing et al., 2021; Xiao et al., 2020).

To improve robustness under practical scenarios where $I^\star$ may undergo transformations, the reverse operation incorporates these transformations $\mathcal{T}$. The extraction process is expressed as:

$$(\hat{I}, \hat{V}) = g_\theta^{-1}(\mathcal{T}(I^\star), z), \tag{14}$$

where $g_\theta^{-1}$ denotes the inverse process of the INN, $z$ is an auxiliary variable that samples from a Gaussian distribution, serving as a complementary component for accurate recovery. Here, $\hat{V}$ is the extracted watermark secrets and $\hat{I}$ approximates the original image $I$. The transformations $\mathcal{T}$ include common operations such as identity (no transformation), Gaussian noise, and JPEG compression. Incorporating these transformations into model training, akin to noisy layers in Zhu et al. (2018), enhances resilience to real-world degradation.

Furthermore, to ensure the embedding process remains imperceptible and preserves the quality of the watermarked image, the visual signature is embedded in the frequency domain using the discrete wavelet transform (DWT). Specifically, the INN takes two inputs: the spatial domain cover image $I$ and the structured spatial secret $V$ (the QR code). Inside the network, $I$ is decomposed into frequency sub-bands. The INN learns a reversible mapping that distributes the information of the spatial QR code $V$ into the frequency features of $I$. This frequency-domain embedding allows the signal to remain imperceptible in the pixel domain while being robustly preserved. During extraction, the INN inverts this transformation, aggregating the distributed frequency signals back into the coherent spatial structure of the original QR code. This approach reduces distortions compared to pixel-domain embedding and enhances the imperceptibility of the watermark.

The INN is trained using a composite loss function designed to simultaneously optimize the recovery performance of the embedded visual signature and preserve the quality of the watermarked image. The loss function is defined as:

$$\mathcal{L}_\theta = \lambda_{\text{emb}}\|I - I^\star\|_2^2 + \lambda_{\text{rec}}\|V - \hat{V}\|_2^2, \tag{15}$$

where $\lambda_{\text{emb}}$ and $\lambda_{\text{rec}}$ are weighting factors balancing the embedding distortion and watermark extraction loss.

Moreover, due to the bijective nature of INNs, any perturbation applied to the watermarked image $I^\star$ directly propagates to the extracted secret, resulting in corresponding artifacts. This effect, observed in Zhang et al. (2024) and illustrated in Fig. 8, reflects the inherent localization property of INNs, which can be leveraged as evidence of tampering.

## 6 Experiments

### 6.1 Experimental Setups

**Datasets and Models.** We use the DIV2K dataset (Agustsson & Timofte, 2017) to train the INN for watermarking embedding and extraction, which contains 800 high-quality 2K resolution images in the training set, and 100 in the validation set. The architecture of INN follows Jing et al. (2021), which has 16 invertible blocks (see Appendix B). For the feature extractor, we leverage a pre-trained image captioning model from Huggingface, which uses ViT model (Dosovitskiy et al., 2021) as the vision encoder and GPT2 (Radford et al., 2019) as the language decoder. For evaluation, we test our method on real images from DIV2K validation dataset, COCO (Lin et al., 2014), and AIGC images generated by stable diffusion (Rombach et al., 2022). The resolution of the images in our experiments is set to 512×512.

**Settings.** We use the QR code encoding and decoding for *PatternEnc* and *PatternDec*, respectively. The QR code of the visual signature in our experiments includes 53×53 modules. For the digital signature

Table 2: Evaluation of image quality and recovery accuracy on DIV2K and COCO datasets.

| | Payload | DIV2K | | | COCO | | |
|---|---|---|---|---|---|---|---|
| | | PSNR↑ | SSIM↑ | *RecAcc* ↑ | PSNR↑ | SSIM↑ | *RecAcc* ↑ |
| HiDDeN | 1× | 37.42 ± 2.57 | **0.987 ± 0.03** | 0.959 ± 0.04 | 37.15 ± 2.78 | **0.988 ± 0.02** | 0.972 ± 0.04 |
| RivaGAN | 1× | **40.43 ± 0.30** | 0.984 ± 0.01 | 0.992 ± 0.04 | **40.54 ± 0.27** | 0.979 ± 0.01 | 0.998 ± 0.02 |
| WAM | 1× | 33.49 ± 1.53 | 0.985 ± 0.01 | 1.000 ± 0.00 | 35.53 ± 1.65 | 0.971 ± 0.01 | 1.000 ± 0.00 |
| DwtDctSvd | 16× | 35.49 ± 3.06 | 0.975 ± 0.01 | 0.994 ± 0.02 | 38.11 ± 2.94 | 0.973 ± 0.01 | 0.999 ± 0.01 |
| MetaSeal (Ours) | **88×** | 34.40 ± 1.97 | 0.965 ± 0.01 | **1.000 ± 0.00** | 34.91 ± 2.31 | 0.963 ± 0.01 | **1.000 ± 0.00** |

algorithm, we use Elliptic Curve Digital Signature Algorithm (ECDSA) with P-256 curve (Hankerson & Menezes, 2021), resulting in a 512-bit signature. For training the INN, the weights for the loss terms are set as $\lambda_{\mathrm{emd}} = 5$ and $\lambda_{\mathrm{rec}} = 1$. We employ the Adam optimizer with an initial learning rate of $1 \times 10^{-4}$. All experiments are conducted on an NVIDIA L40S GPU.

**Comparison Methods.** It is worth noting that there exists a vast array of image watermarking methods, making exhaustive comparison infeasible. Therefore, we focus on comparing with several representative/SOTA methods, including DwtDctSvd (Cox et al., 2007) (used in the official Stable Diffusion model), HiDDeN (Zhu et al., 2018), RivaGAN (Zhang et al., 2019), WAM (Sander et al., 2025), and Stable Signature (Fernandez et al., 2023). The first four methods are post-hoc techniques that can be applied to any image, whereas Stable Signature is specifically designed for generative images, embedding the watermark during image generation. It is important to note that each method supports a different payload capacity and it cannot be adjusted to the same large payload as MetaSeal without compromising its effectiveness. The limitations of other watermarking techniques have been discussed in the Sec. 8.

**Evaluations** We use the Peak Signal-to-Noise Ratio (PSNR) and Structural Similarity Index (SSIM) to measure the quality of the watermarked image (Hore & Ziou, 2010). In addition, recovery accuracy (*RecAcc*) quantifies the percentage of successfully recovered watermark secrets. For comparison methods that all use bit strings as secrets, it is calculated as:

$$RecAcc_{bit} = \frac{1}{N} \sum_{i=1}^{N} \mathbb{I}(b_i = \hat{b}_i) \times 100\%, \tag{16}$$

where $\hat{b}_i$ is the decoded bit and $b_i$ is the ground truth. For MetaSeal, the secret is visual structured pattern, where the recovery accuracy is calculated as:

$$RecAcc_{pattern} = \frac{1}{H \times W} \sum_{i=1}^{H} \sum_{j=1}^{W} \mathbb{I}(I_{i,j} = \hat{I}_{i,j}) \times 100\%, \tag{17}$$

where $H \times W$ is the pattern resolution. We also report verification accuracy (*VerAcc*), which reflects whether the extracted secret can correctly confirm image attribution. For our method, which embeds cryptographic signatures, *VerAcc* is defined as the proportion of successfully verified signatures, requiring exact recovery to satisfy cryptographic validation.

## 6.2 Balance Between Payload and Accuracy

To demonstrate MetaSeal achieves a good trade-off between payload size and recovery/verification accuracy, we evaluate its performance on diverse datasets and compare it with multiple watermarking methods, as shown in Tab.2 and Tab.3. Existing approaches, such as HiDDeN, use fixed-length bitstreams (typically <100 bits), resulting in minimal payloads. Specifically, in our experiments, HiDDeN embeds a 32-bit secret in a $512 \times 512$ image, yielding a payload of $\sim 1 \times 10^{-4}$ bit per pixel, which we normalize as the baseline for comparison.

In contrast, MetaSeal introduces a paradigm-shifting approach by designing visual signatures encoded in QR codes. Our payload calculation, defined as the ratio of embedded modules to total image pixels, demonstrates an 88× increase compared to HiDDeN, RivaGAN, and WAM. As shown in Fig. 4, MetaSeal achieves the

Table 3: Evaluation of image quality and recovery accuracy on AIGC dataset.

|  | Payload | PSNR ↑ | SSIM ↑ | *RecAcc* ↑ |
|---|---|---|---|---|
| HiDDeN | $1\times$ | $33.79 \pm 3.36$ | $\mathbf{0.989 \pm 0.01}$ | $0.966 \pm 0.04$ |
| RivaGAN | $1\times$ | $\mathbf{40.65 \pm 0.23}$ | $0.981 \pm 0.01$ | $0.966 \pm 0.06$ |
| WAM | $1\times$ | $34.04 \pm 1.77$ | $0.986 \pm 0.01$ | $\mathbf{1.000 \pm 0.00}$ |
| Stable Signature | $1.5\times$ | $27.81 \pm 2.26$ | $0.916 \pm 0.03$ | $0.982 \pm 0.03$ |
| DwtDctSvd | $16\times$ | $34.95 \pm 2.83$ | $0.973 \pm 0.01$ | $0.994 \pm 0.02$ |
| MetaSeal (Ours) | $\mathbf{88\times}$ | $34.43 \pm 2.94$ | $0.965 \pm 0.01$ | $\mathbf{1.000 \pm 0.00}$ |

best secret recovery accuracy at the largest payload, whereas other methods experience a low recovery accuracy as the payload increases. For HiDDeN, the recovery accuracy drops from 0.987 to around 0.5 when secret message length increases from 32 bits to 512 bits, as shown in Fig. 2. Moreover, MetaSeal consistently delivers perfect secret extraction, as quantified by *RecAcc*. This accurate recovery enables perfect verification accuracy for non-transformed watermarked images, as shown in Fig.5, where verification accuracy drops drastically for DwtDctSvd under the same payload.

Notably, this increased payload capacity does not significantly compromise image quality. MetaSeal surpasses Stable Signature's performance on AIGC datasets and achieves comparable quality to DwtDctSvd on DIV2K, a method already integrated into Stable Diffusion's image watermarking system (Rombach et al., 2022). Also, its image quality is slightly better than WAM on most cases, which can also achieve perfect recovery accuracy but only supports a small payload. The visual performance of MetaSeal for both embedding and extraction is demonstrated in Fig. 6.

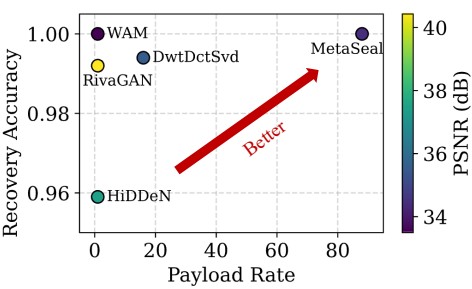

Figure 4: MetaSeal achieves the best performance between payload and recovery accuracy on DIV2K.

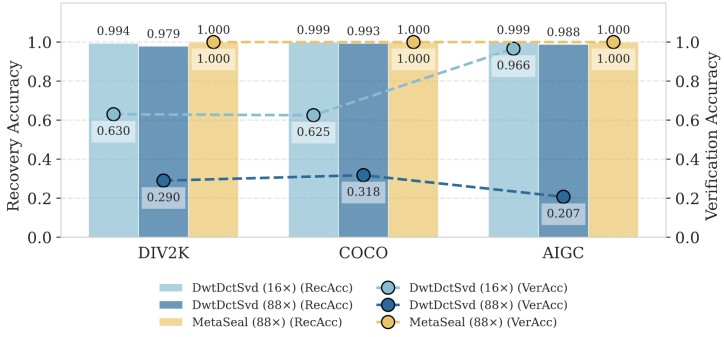

Figure 5: As the payload increases for DwtDctSvd, its *VerAcc* drops significantly, while MetaSeal maintains perfect recovery and verification accuracy.

**Takeaway 1:** By leveraging the inherent resilience of structured visual signatures and invertible embedding strategies, MetaSeal achieves a superior balance between payload capacity and recovery/ verification accuracy, while maintaining image quality comparable to real-world watermark implementation.

## 6.3 Robustness against Benign Transformations

As discussed in Sec. 2.2, robustness against watermark removal attacks, including transformations or adversarial modifications that could erase the watermark, is not a primary objective for attribution watermarks. *In fact, such robustness implies tolerance to modifications, which contradicts the goal of attribution watermarking: to signal tampering and invalidate attribution when meaningful changes occur.* Therefore, unlike prior methods that prioritize resistance to editing and removal, MetaSeal is designed to remain robust only under *benign transformations*. We also provide its performance under removal attacks in the Appendix C.

To ensure attribution survives standard usage, Fig. 7 quantifies verification accuracy under five types of transformations: brightness adjustment, contrast variation, blurring, Gaussian noise, and JPEG compression (with scaling and cropping in Appendix D). MetaSeal maintains stable verification accuracy within tolerance

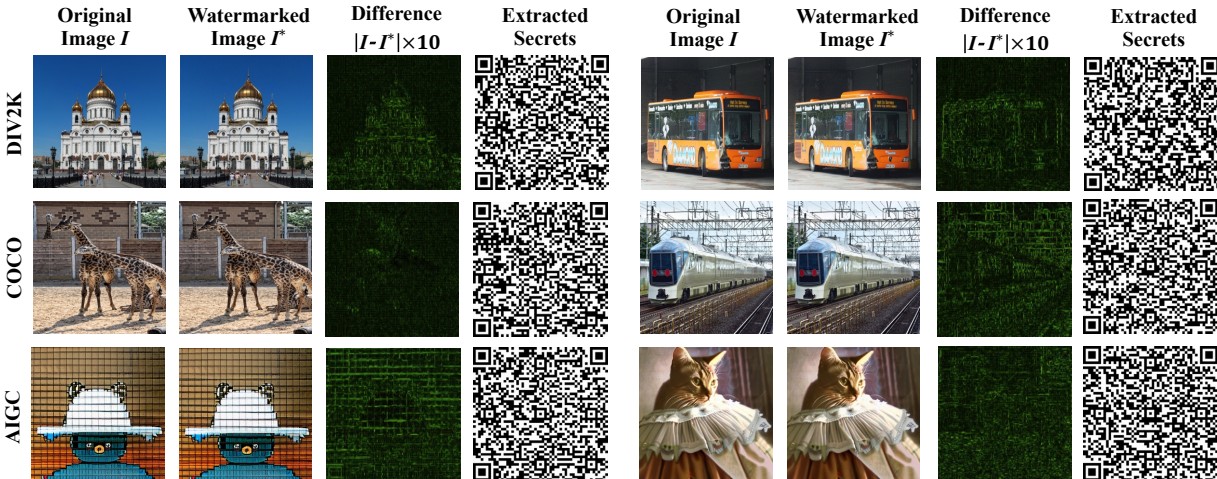

Figure 6: The visual performance of `MetaSeal` for both embedding and extraction across different datasets.

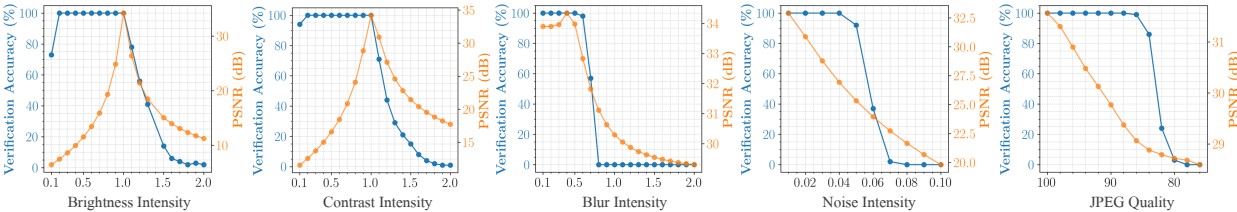

Figure 7: `MetaSeal` performance under varying distortion intensities. The red star denotes the performance when the transformation is not added (baseline). Each panel plots verification accuracy (blue line, left axis) and the cover image PSNR (orange line, right axis) alongside the distortion intensity for brightness, contrast, blur, noise, and JPEG compression.

thresholds: blur deviation $\sigma < 0.7$, noise variance $< 0.05$, and JPEG factor $> 84$. It is highly resilient to brightness reduction (down to 20%) and contrast reduction (10%) but is sensitive to excessive enhancement beyond 110% of baseline. This sensitivity arises because brightness enhancement amplifies high-frequency noise, which compromises watermark recovery.

Notably, `MetaSeal` maintains high verification accuracy (blue curves) when distortion magnitudes remain low, even as these transformations degrade the visual quality of the watermarked images (orange curves). We further visualize `MetaSeal`'s performance under these transformations, as well as horizontal flipping, in Appendix D. As shown in Fig. 7, transformations may introduce artifacts in extracted secrets, but QR codes remain accurately decodable.

This robustness stems primarily from the design of the visual signature and the trained INN. The visual signature, converted into meaningful patterns like QR codes, acts as a specialized encoding scheme that can tolerate minor errors, enhancing recovery reliability. Meanwhile, the INN enhances robustness to benign transformations by embedding the signature in the frequency domain and incorporating noise layers during training, leading to minimal impact due to moderate benign transformations. Together, these components enable our scheme to support reliable verification even when images undergo typical, non-malicious transformations.

## 6.4 Sensitivity to Perturbations

When watermarked images undergo malicious perturbations, such as image editing, `MetaSeal` demonstrates an intentional sensitivity by reflecting these changes in the recovered visual signature. As shown in Fig. 8,

editing the watermarked image results in corresponding artifacts in the recovered visual signature. This correlation occurs because the embedded visual signature experiences the same editing operations as the image itself, making modifications visibly detectable. Besides, we also observe that the degree of tampering impacts decoding accuracy: major edits (e.g., introducing more than 10% pixel-level changes) often result in decoding failure. Interestingly, when the extracted QR code exhibits partially intact square modules—suggesting successful embedding by the oracle INN but subsequent manipulation—this can serve as visual evidence of tampering. In our design, tampering artifacts serve as an interpretability feature assessed by human inspection, but could be further leveraged by training a binary classifier for automatic detection.

The ability to reveal pixel-level changes tied to specific perturbations via INN enables localization of tampering, which is also demonstrated in Zhang et al. (2024). This characteristic highlights `MetaSeal`'s dual advantages: it not only verifies attribution but also provides concrete evidence of tampering attempts. This represents a significant advancement over traditional watermarking methods, which typically embed fixed bits and produce only binary verification results.

> **Takeaway 2:** `MetaSeal` resolves the tension between content dependency and robustness by embedding a semantic-aware visual signature—uniquely tied to image content—using an invertible network trained with frequency-domain embedding and noise augmentation. This design ensures that the watermark remains stable under benign transformations yet fragile to malicious perturbations that alter content semantics, achieving reliable attribution without sacrificing tamper sensitivity.

### 6.5 Anti-forgery Demonstration

### 6.5.1 Current Forgery Attacks

We demonstrate that `MetaSeal` is resilient to a broad range of forgery attacks, including current forgery attacks discussed in Sec. 3.1. For replay attacks, even if attackers know the detection mechanism, they cannot forge valid signatures due to the use of asymmetric cryptography. Specifically, the secrets in `MetaSeal` are signed with a private key securely held by the model owner or content creator. As a result, attackers cannot generate valid signatures for unrelated images. Moreover, mixup attacks rely on estimating a ground truth watermark by aggregating residuals from multiple watermarked images. However, in `MetaSeal`, the watermark is content-dependent and varies across images. This prevents attackers from estimating a valid signature for a specific target image, rendering such attacks ineffective.

For PGD attacks, `MetaSeal` avoids the weakness of binary detectors that directly classify images as watermarked, which makes them susceptible to adversarial perturbations (Saberi et al., 2024). Instead, `MetaSeal` uses an INN only to reconstruct the embedded secret, while verification is performed through cryptographic validation of the recovered secret. Thus, the INN is merely a reconstruction tool, and the secret—bound to the private key—ultimately decides validity. Since attackers lack the private key, they cannot generate valid secrets, making adversarial attacks effective against CNN-based detectors ineffective against our scheme.

Moreover, recent works have proposed forgery attacks targeting diffusion-based, content-agnostic watermarks that embed fixed or model-specific signals (Müller et al., 2025; Jain et al., 2025), enabling watermark estimation, cancellation, or forgery from limited observations. These attacks are not directly applicable to `MetaSeal` due to fundamental differences in watermark design. In contrast, `MetaSeal` embeds content-dependent cryptographic signatures, such that each image carries a unique signed message; consequently, attacks that rely on reusing or estimating a fixed watermark signal do not directly transfer. This distinction highlights an inherent limitation of prior watermark designs and motivates the need for attribution mechanisms tailored to forgery-resistant settings.

### 6.5.2 Adaptive Forgery

We consider a stronger adversary who obtains a valid visual signature $\hat{V}$ from a watermarked image $I^\star$ and attempts to transplant it into a different image $I_d$ with similar content, i.e., $f(I^\star) \approx f(I_d)$. It reflects an

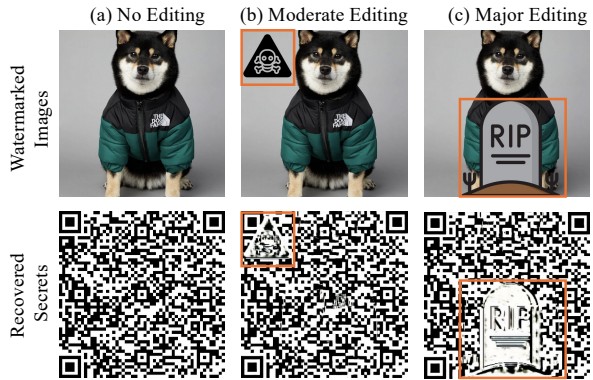

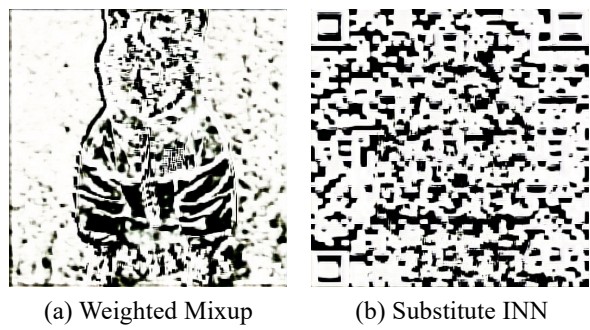

Figure 8: Sensitivity of `MetaSeal` to editing. The recovered secrets provide visual tampering evidence.

Figure 9: Robustness to adaptive forgery. Attacks fail to produce decodable secrets.

adaptive threat model in which the attacker maintains semantic consistency to evade content-dependent checks.

**Weighted Mixup.** In this attack, the adversary blends the extracted visual signature $\hat{V}$ into an unmarked image $I_d$ via a weighted mixup. Since a successful attack must maintain an imperceptible watermark, we apply a weight of 0.05. The extracted secrets from the forged image are shown in Fig. 9(a), where the oracle INN fails to recover a valid QR code, demonstrating the ineffectiveness of this attack.

**Substitute INN.** We further consider an even stronger threat: an attacker trains a substitute INN using access to the full training dataset, including original images and their corresponding secrets. The attacker then uses this substitute model to embed $\hat{V}$ into a new image $I_d$. As shown in Fig. 9(b), the extracted signature from the forged image fails verification—producing an unreadable QR code. This failure occurs because `MetaSeal` relies on invertibility between embedding and extraction paths, which only holds when both use the same trained weights. Due to training randomness and architectural differences, the substitute INN cannot replicate the oracle INN's embedding behavior.

> **Takeaway 3:** `MetaSeal` ensures anti-forgery security through: **1) Cryptographic Binding:** Forgery without access to the private key `sk` is computationally infeasible due to the security of ECDSA. **2) Content Dependency:** The watermark is semantically bound to the image content $f(I)$, preventing reuse across different images. **3) Invertibility Isolation:** The INN-based embedding is non-replicable due to model-specific parameters, mitigating substitute model attacks.

## 7 Discussion

### 7.1 Impact of Semantic Granularity

To develop content-dependent watermarks, we leverage semantic extraction, where the level of granularity influences the trade-off between security and robustness to benign transformations. A highly detailed semantic description significantly reduces the risk of watermark reuse, enhancing security. However, increased detail also leads to a larger payload, increasing the density of QR codes and making accurate watermark recovery more challenging after benign transformations. Conversely, if the semantic representation is too simple, e.g., only identifying the main object, the watermark may remain valid for other images containing the same object, increasing the risk of false positives.

We conducted experiments to analyze how semantic granularity affects watermark robustness through its impact on QR code density. Our baseline implementation uses a QR code module of $53 \times 53$. When the semantic description becomes more fine-grained, incorporating additional image details, the required payload increases, necessitating denser QR codes. To quantify this effect, we tested a denser QR code configuration

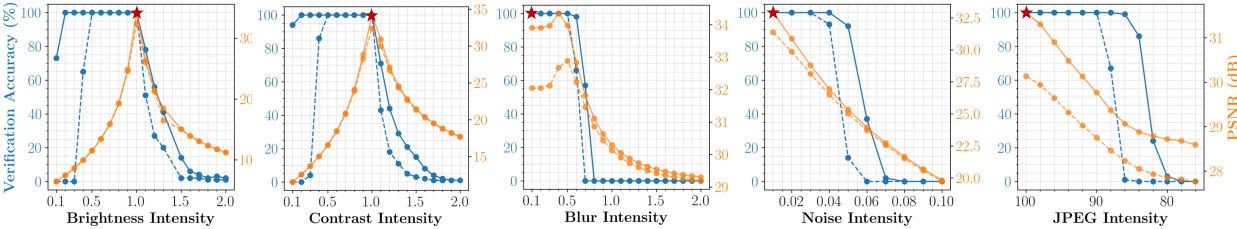

Figure 10: `MetaSeal` performance under varying distortion intensities with different payload. The solid line denotes a small payload while the dotted line denotes a large payload.

($85 \times 85$ modules) that accommodates more detailed semantic descriptions. As illustrated in Fig. 10, while the denser QR code maintains perfect verification accuracy for unmodified images, it shows reduced resilience to benign transformations compared to the sparser baseline configuration.

## 7.2 Scalability and Efficiency

Our method is resolution-agnostic and can be applied to images of varying resolutions without retraining. As shown in Table 4, `MetaSeal` achieves perfect recovery and verification accuracy across all tested resolutions. Notably, both PSNR and SSIM improve as the image resolution increases. This trend arises because the embedded secret QR code has a fixed number of modules, making the relative payload smaller in larger images. As a result, the watermark introduces less perceptual distortion, enhancing image quality while maintaining exact signature reconstruction. Moreover, we quantify the computational overhead of watermarking by measuring embedding and verification times across different image resolutions (Ta-

Table 4: Performance across different resolutions

| Dataset | Metric | 256×256 | 1024×1024 | 2048×2048 |
|---------|--------|---------|-----------|-----------|
| DIV2K | PSNR | 32.13 | 36.89 | 39.08 |
| | SSIM | 0.9342 | 0.9772 | 0.9845 |
| | *RecAcc* | 1.000 | 1.000 | 1.000 |
| | *VerAcc* | 1.000 | 1.000 | 1.000 |
| COCO | PSNR | 32.58 | 38.54 | 40.69 |
| | SSIM | 0.9315 | 0.9821 | 0.9890 |
| | *RecAcc* | 1.000 | 1.000 | 1.000 |
| | *VerAcc* | 1.000 | 1.000 | 1.000 |
| AIGC | PSNR | 30.46 | 39.00 | 40.50 |
| | SSIM | 0.9293 | 0.9736 | 0.9774 |
| | *RecAcc* | 1.000 | 1.000 | 1.000 |
| | *VerAcc* | 1.000 | 1.000 | 1.000 |

ble 5). As shown in the results, both embedding and verification times increase with resolution, reflecting the higher computational cost of processing larger images. Embedding time increases more sharply than verification time, primarily due to the computational cost of semantic extraction and visual signature generation using the private key. However, embedding and verification remain efficient even at high resolutions (e.g., <1s for one 1024×1024 image), demonstrating the practicality of `MetaSeal` for real-world implementation.

## 7.3 Limitations and Future Works

While `MetaSeal` provides strong security guarantees through cryptographic signatures and content-dependent design, several open challenges remain, leaving room for improvement. One limitation is the relatively large payload. Unlike prior methods that embed short binary messages (e.g., 32 bits), `MetaSeal` uses a

Table 5: Watermarking computational overhead: Embedding and verification time (seconds) per batch (16 images) across resolutions.

| Resolution | 256×256 | 512×512 | 1024×1024 | 2048×2048 |
|------------|---------|---------|-----------|-----------|
| **Embedding (s)** | 1.291 | 1.515 | 3.667 | 15.198 |
| **Verification (s)** | 0.028 | 0.184 | 0.861 | 9.289 |

cryptographic signature, typically several hundred bits in length. This increased payload can sometimes slightly degrade visual quality, as reflected in lower PSNR and SSIM scores compared to methods optimized for imperceptibility. The use of high-contrast black-and-white QR codes further contributes to visible artifacts. Another factor is the optimization of robustness in the INN through the use of noisy layers. While this improves resilience to benign transformations such as compression and blurring, it leads to watermarks

being embedded primarily in the mid-frequency domain. Although this design enhances robustness, it also degrades PSNR compared to high-frequency embedding, which—while less visible—is too fragile for reliable attribution (see Appendix E). This reflects an inherent trade-off between robustness and imperceptibility. A promising direction for future work is to explore alternative visual encoding schemes. For example, adopting softer patterns or low-contrast colored codes could reduce visual artifacts while maintaining the robustness and decodability of the embedded signature. Another limitation is that the current robustness to benign transformations is limited. Improving robustness to benign transformations inevitably compromises invisibility and increases overall tolerance, which may expand the set of modified images that still verify and thus raise the risk of misattribution. How to achieve selective robustness, i.e., strong robustness to certain benign transformations while remaining sensitive to others, remains an important and unresolved challenge for future work.

## 8  Related Works

Recent efforts have explored content-dependent watermarking for checking image authenticity. For example, Evennou *et al.*(Evennou et al., 2024) encode a semantic textual description of the image and verify authenticity by comparing it with decoded semantics. Similarly, Arabi *et al.* (Arabi et al., 2025) embed text prompts into diffusion-generated images by perturbing the initial noise, and then compare the recovered noise with image semantics for verification. While both methods address image authenticity, they require access to the secret key at verification time, limiting them to private or semi-trusted settings without public verifiability. Besides, there are a few works that have addressed forgery attacks in specific domains. Bileve (Zhou et al., 2024) targets spoofing in language watermarking, but is not applicable to image content. Methods like (Gunn et al.) propose watermarking strategies specifically for diffusion-generated images, offering limited generality for real-world photographs or other generative models. EditGuard (Zhang et al., 2024) improves tamper detection using dual watermarks—one for binary verification and one for localization—embedded via INNs. However, both are fixed and image-agnostic, making them susceptible to cross-image forgery and lacking semantic binding. WAM (Sander et al., 2025) focuses on localization with a high recovery rate but is limited to 32-bit payloads. Neither approach supports high-bit payloads or cryptographic attribution. While Gaussian Shading (Yang et al., 2024b) incorporates cryptographic techniques, its goal is distribution-preserving sampling using ChaCha20 ciphers. It does not address attribution, forgery resistance, or public verification, because it embeds fixed, random watermark bits rather than content-dependent attribution information, and relies on symmetric cryptographic primitives, which require keeping the secret key private. As a result, the verification process cannot be made publicly verifiable without compromising security.

Other closely related lines of work include fingerprinting, steganography, and watermarking with INNs. Fingerprinting approaches (Yu et al., 2022; Kim et al., 2024) achieve attribution by modifying model weights so that generated images exhibit identifiable characteristics, rather than embedding secrets directly into images, and typically support only limited payloads ($\sim$128 bits). Steganography methods (Jing et al., 2021; Lu et al., 2021) prioritize covert communication, emphasizing large payloads and resistance to steganalysis; however, they are typically fragile even to benign transformations and do not aim to support attribution or verification. INN-based watermarking methods such as Ma et al. (2022) and EditGuard Zhang et al. (2024) leverage invertibility for improved extraction or tamper localization, but embed fixed, content-agnostic bit strings, making them vulnerable to replay and cross-image forgery.

In summary, while prior work has made progress in isolated aspects—such as content-dependence, robustness, or tamper detection—`MetaSeal` is the first to integrate semantic binding, cryptographic verification, public verifiability, and visual signature design into a unified framework.

## 9  Conclusion

This paper presents `MetaSeal`, a reliable and cryptographically verifiable framework for image attribution. Unlike prior watermarking methods that rely on fixed patterns or detector-based verification, `MetaSeal` introduces content-dependent signatures encoded as structured visual patterns and embedded using invertible neural networks. This design achieves three key goals: binding attribution to image semantics, enabling

exact signature recovery, and ensuring public verifiability without metadata. Our empirical results demonstrate that `MetaSeal` scales to high-capacity payloads while maintaining perfect verification accuracy and strong image quality. It remains robust to benign transformations yet sensitive to malicious edits, providing not only forgery resistance but also visual evidence of tampering. Moreover, `MetaSeal` withstands adaptive attacks, benefiting from the cryptographic unforgeability of signatures and the non-replicability of invertible embedding. By integrating semantic binding, cryptographic security, and structured visual encoding, `MetaSeal` offers a practical and provable defense against image misattribution via forgery attacks.

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

# A  Analysis of Structured Visual Embedding

This section provides an intuitive and probabilistic analysis of why structured visual embedding is better suited for cryptographic attribution than conventional bit-wise watermarking. We emphasize that the analysis is not a formal security proof, but a justification of the design choice underlying our Visual Attribution Signature.

**Limitations of bit-wise embedding.** Consider a cryptographic signature $S \in \{0,1\}^N$ with $N = 512$. In bit-wise watermarking, correct verification requires near-perfect recovery of all bits (Cox et al., 2007; Zhu et al., 2018; Fernandez et al., 2023). Let $p_e$ denote the per-bit error probability under benign image transformations. The probability of successful extraction is

$$P_{\text{bit}}(\text{success}) = \prod_{i=1}^{N} P(\hat{b}_i = b_i) = (1 - p_e)^N.$$  (18)

Even for a modest error rate $p_e = 0.01$, this yields $(0.99)^{512} \approx 0.006$, demonstrating that bit-wise embedding is intrinsically fragile when scaled to cryptographic payload sizes.

**Structured visual embedding.** Instead of treating the signature as independent bits, our method represents it as a *structured visual pattern* $V \in \{0,1\}^{h \times w}$ with explicit spatial organization and redundancy. This converts extraction from a fragile bit-recovery problem into a pattern-decoding problem, yielding two key advantages.

**1) Local redundancy via spatial structure.** Due to structural regularity, neighboring elements in $V$ are statistically correlated. Let $\mathcal{N}(i)$ denote the spatial neighborhood of element $i$. Then

$$P(\hat{b}_i = b_i \mid \mathcal{N}(i)) > P(\hat{b}_i = b_i),$$  (19)

reflecting the fact that local consistency can be exploited during decoding. This form of redundancy is *inherent to the visual structure* and does not arise in independent bit-wise embedding.

**2) Global error tolerance via structured decoding.** In our implementation, the visual pattern instantiates a QR-style code with Reed–Solomon (RS) error correction. Let $(n, k, d)$ denote the RS parameters, with error-correction capability $t = \lfloor (d-1)/2 \rfloor$. Assuming a symbol error rate $p_s$, the probability of successful decoding is

$$P_{\text{pattern}}(\text{success}) = \sum_{i=0}^{t} \binom{n}{i} p_s^i (1 - p_s)^{n-i}.$$  (20)

Unlike bit-wise decoding, failure occurs only when errors exceed a global threshold, yielding graceful degradation rather than catastrophic failure.

While error-correcting codes can in principle be applied to any fingerprint, bit-wise watermarking still requires *exact recovery of the encoded bitstream* from noisy image features, which remains the dominant failure mode. In contrast, structured visual embedding integrates redundancy, spatial correlation, and decoding geometry *at the visual level*, enabling reliable recovery even when individual elements are corrupted. This distinction is critical for cryptographic verification, which tolerates no bit errors. This analysis explains why structured visual embedding fundamentally alters the robustness–capacity trade-off, making large, cryptographically meaningful payloads feasible. It provides the design rationale for `MetaSeal`'s visual attribution signature, rather than a formal cryptographic guarantee.

## B  Invertible Neural Networks

The architecture of INN has 16 invertible blocks. Each of them are composed of three modules: $\phi(\cdot)$, $\rho(\cdot)$, and $\eta(\cdot)$. These modules are built by dense blocks, which has better representation ability than convolutional blocks and residual blocks, as demonstrated in Jing et al. (2021). In particular, iven an input feature map $\mathbf{x} \in \mathbb{R}^{C \times H \times W}$, the block consists of five convolutional layers. Each of the first four layers has 32 output channels and is followed by a LeakyReLU activation. The input to each convolutional layer is the concatenation of the original input $\mathbf{x}$ and all preceding intermediate features, forming progressively richer

Watermarked image     Recovered secret       With regenerate attack     Recovered secret

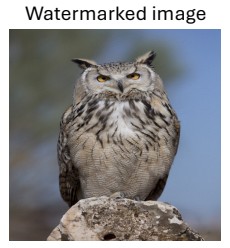 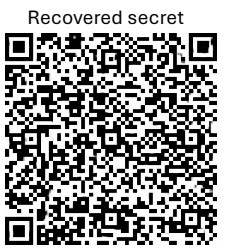 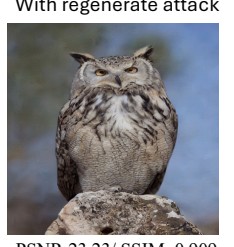 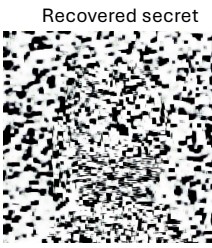

PSNR:35.58/ SSIM: 0.965               PSNR:23.23/ SSIM: 0.909

Figure 11: **Left:** Image watermarked with `MetaSeal` and its recovered secret. **Right:** Apply a regeneration attack (Zhao et al., 2024) to the watermarked image and its recovered secret.

representations. Formally, the block can be described as:

$$\mathbf{f}1 = \text{LeakyReLU}(\text{Conv}_{3\times3}(\mathbf{x}))$$
$$\mathbf{f}2 = \text{LeakyReLU}(\text{Conv}_{3\times3}([\mathbf{x}, \mathbf{f}_1]))$$
$$\mathbf{f}3 = \text{LeakyReLU}(\text{Conv}_{3\times3}([\mathbf{x}, \mathbf{f}_1, \mathbf{f}_2]))$$
$$\mathbf{f}4 = \text{LeakyReLU}(\text{Conv}_{3\times3}([\mathbf{x}, \mathbf{f}_1, \mathbf{f}_2, \mathbf{f}3]))$$
$$\mathbf{y} = \text{Conv}_{3\times3}([\mathbf{x}, \mathbf{f}_1, \mathbf{f}_2, \mathbf{f}_3, \mathbf{f}_4])$$

Here, $[\cdot]$ denotes channel-wise concatenation. The final output $\mathbf{y}$ has the same number of channels as the desired output dimension. To stabilize training, the weights of the final convolution layer are initialized to zero, ensuring the block initially behaves like an identity mapping.

The INN used in `MetaSeal` is lightweight, with only 4 million parameters, making it computationally efficient and easy to integrate into existing systems. Its compact size enables seamless deployment in resource-constrained environments and allows it to be embedded into image generation pipelines for real-time watermarking with minimal overhead. This efficiency distinguishes it from larger encoder-decoder models and makes it suitable for scalable applications such as online content creation or platform-level attribution enforcement.

## C    Performance under Regeneration Attack

Here we evaluate the behavior of `MetaSeal` under advanced watermark removal attacks (Zhao et al., 2024; Liu et al., 2025). Specifically, Fig. 11 reports results under a regeneration attack based on diffusion models (Zhao et al., 2024). After regeneration, the recovered visual signature is no longer scannable, and attribution verification fails. This outcome is expected: Regeneration fundamentally rewrites the image through an iterative diffusion process, which disrupts the invertible embedding structure learned by the INN. As a result, the extracted secret is no longer intact and fails cryptographic verification.

We further observe that successful removal via regeneration comes at a substantial perceptual cost. Compared to the unwatermarked image, the original watermarked image has PSNR 35.58 dB and SSIM 0.965, whereas the regenerated image degrades to 23.23 dB PSNR and 0.909 SSIM. This degradation indicates that the attack trades attribution removal for a significant loss in image fidelity, reducing the practical value of the manipulated content.

## D    More Results of Transformations

Here we add more evaluations against common transformations, as shown in Fig. 12. For scaling, the verification accuracy remains perfect (100%) across scale factors of 2.0, 1.5, and 1.0, demonstrating strong resilience to upscaling. However, accuracy drops when the scale is reduced further. This indicates that excessive downscaling can severely distort or erase the watermark signal. For random cropping, verification accuracy exhibits gradual degradation as the crop ratio increases. Starting from 96.97% at a 5% crop, the

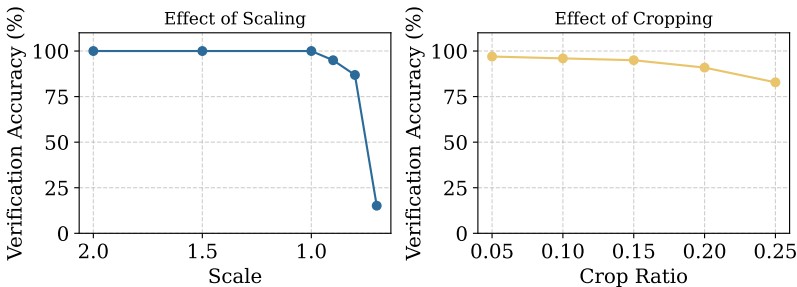

Figure 12: The resilience of `MetaSeal` against scaling and random cropping.

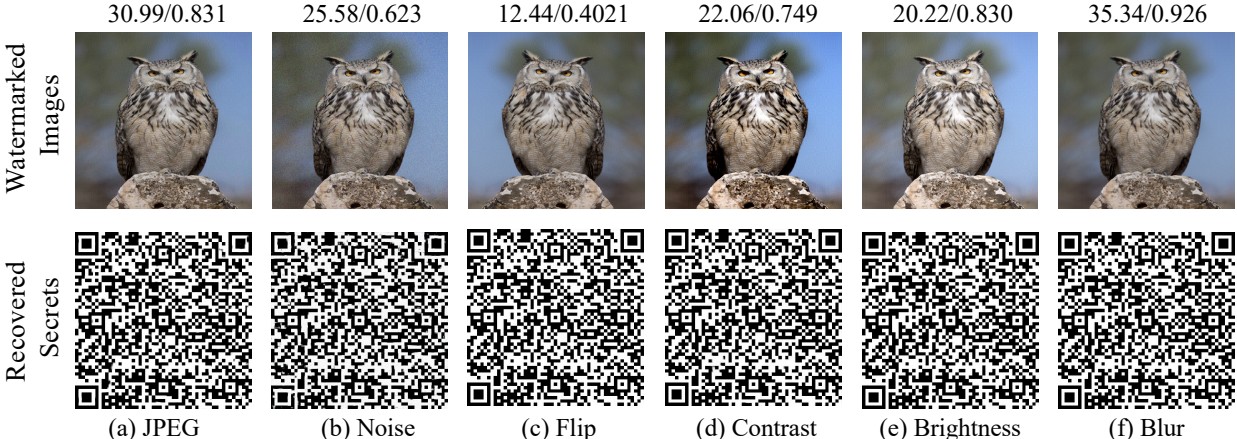

Figure 13: Evaluation of secret recovery from watermarked images under benign Transformations. The PSNR/SSIM is displayed for each image compared with the unwatermarked image. The top row shows watermarked images subjected to different transformations, while the bottom row displays the recovered secrets (QR codes) under each transformation. One can scan these QR codes to test the consistency of the recovered secrets.

accuracy declines to 82.83% at a 25% crop. These results suggest that the watermark maintains robustness under moderate cropping but becomes vulnerable when a substantial portion of the image is removed.

Furthermore, we visualize the recovered secrets from watermarked images subjected to various transformations. We apply a set of transformations with detailed parameter settings provided in Tab. 6. The visualization results, shown in Fig. 13, demonstrate that our method consistently retrieves accurate information from the extracted QR codes, even when transformations degrade the quality of the recovered secrets. Notably, despite distortions such as compression and noise, the encoded information remains decodable, highlighting the resilience of our approach against common image modifications. This robustness ensures that our watermarking system remains reliable in practical deployment scenarios where images undergo benign processing or distribution-related alterations.

Table 6: Transformation parameters.

| Transformation | Parameter |
|---|---|
| Flip | Horizontal Flip |
| Brightness | `brightness_factor=1.2` |
| Contrast | `contrast_factor=1.5` |
| Gaussian Blur | `kernel_size=3`, `sigma=0.5` |
| Gaussian Noise | `mean=0`, `std=1`, `intensity=0.05` |
| JEPG | `Q=90` |

## E    Effect of Noisy Layers

We examine the effect of training the INN model with noisy layers. Fig. 14 compares the frequency-domain decomposition of watermarked images produced with and without noisy layers during training. When noisy

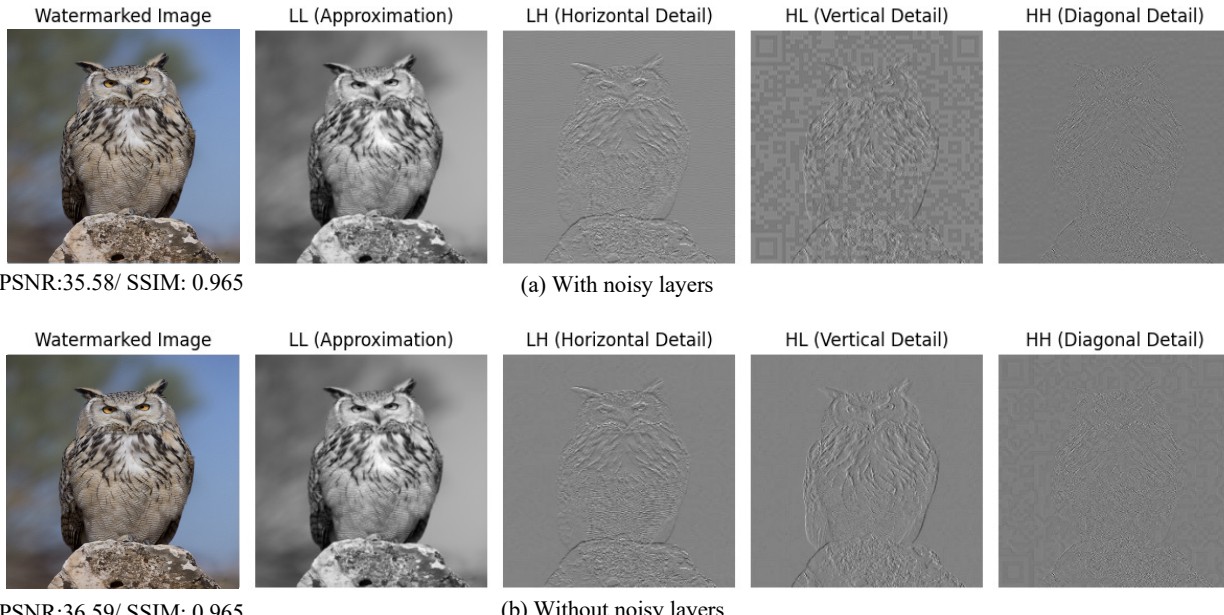

Figure 14: Wavelet decomposition of watermarked images (a) with and (b) without noisy layers during training. Without noisy layers, the watermark tends to concentrate in the high-frequency bands (HH), making it more imperceptible but fragile. In contrast, noisy layers promote embedding in the mid-frequency bands (particularly HL), enhancing robustness to benign transformations at the cost of slightly increased perceptibility (lower PSNR).

layers are used, watermark energy is more prominently concentrated in the mid-frequency bands (especially the HL component), which enhances robustness to common image transformations such as compression. In contrast, without noisy layers, the watermark is more uniformly distributed or biased toward high-frequency regions (HH), which may improve imperceptibility but leads to poor resilience under benign distortions. Specifically, it cannot resist compression with any factors, i.e., with even a bit compression (Q=99), the verification accuracy will drop to 0. This fragility highlights a critical limitation: while high-frequency embedding may yield imperceptible watermarks, it fails to survive even the slightest benign transformation. In real-world scenarios, robustness to benign transformations is essential for reliable attribution. This observation illustrates the inherent trade-off between robustness and invisibility. In our implementation, the PSNR dropped a little bit after adding noisy layers to improve robustness. However, this degradation is barely observed by human perceptions, while the resulting improvement in robustness against benign transformations is substantial.

