# OpenReview forum: "MetaSeal: Defending Against Image Attribution Forgery Through Content-Dependent Cryptographic Watermarks"
_TMLR — Accepted by TMLR_

### Review · Reviewer_pFdn · 2025-11-23

**Summary Of Contributions:**

This paper suggests a publicly verifiable watermark that is attached to the content of the watermarked images. They embed the watermark using a discrete wavelet transform (DWT). An INN (Invertible Neural Network) is used to embed the image together with the watermark in a way that will be later detectable, with some robustness to removal transformations. After recovery of the embedded signal, in the form of a QR code, the watermark is validated in terms of spatial integrity and relation to the content of the images. Empirical experiments validate the watermark's robustness to removal attacks and reasonable robustness to forgery attacks.

Main Strengths:
1. The problem of making a publicly verifiable watermark that is attached to the content is interesting, and important.
2. The breakdown of watermark usages to Copyright protection, Authentication, Attribution, is nice. I also liked Sec.3.2.
3. Empirical evaluation validates the claim about resistance to some forgery and removal attacks.

Main weaknesses:
1. Reading the paper end to end, a lot of the details remain unclear to me. It was not clear to me what about the INN make the watermark robust (unlike "regular"  adversarial training)
2. While the paper does have merit, imprecise use of language often results in exaggerated claims.
3. No GitHub code is attached for reproducibility purposes

**Additional Comments:**

1. What "IWT" stands for? (Figure 3).
2. Claims such as "We propose MetaSeal, the first image attribution watermark that provides innovative solutions to these fundamental questions" are not accurate (as can also be seen by the works brought by the authors in Sec. 8)
3. For citations, please use the conference /  journal version of each cited paper when available (rather than the arXiv one)
4. Some sentence are unclear, E.g.:
- "To realize this, we implement f using an image-to-text model that maps visual content to a deterministic textual description. Specifically, we decompose f into an encoder–decoder architecture” (not clear, especially the role of the text there).
- Most of Sec. 5.4
- "While Gaussian Shading (Yang et al., 2024b) incorporates cryptographic techniques, its goal is distribution-preserving sampling using ChaCha20 ciphers. It does not address attribution, forgery resistance, or public verification” - why it does not address attribution?
5. "To address these challenges, MetaSeal introduces two key innovations. First, it generates content-dependent signatures that encapsulate the image’s semantic information, ensuring the watermark is tied to the content rather than pixel details.". Again, while this work has some merit, none of this parts are truly novel w.r.t. to pervious methods (and not even the combination of these)
6. Fig. 3 can be made much clearer. E.g., what is the loss and function of the INN there?
7. It is not clear how the very clean QR codes (e.g., Fig.6) are related to the DWT embedding.
8.  Claims such as “Forgery without access to the private key sk is computationally infeasible due to the security of ECDSA” ; “The INN-based embedding is non-replicable due to model-specific parameters, stopping substitute model attacks. “- are too strong, especially when works such as [2,3] are not evaluated (let alone new attacks not yet invented).
9. Figure 11 - the left plot is not clear to me. The closer the scale is to 1 the lower the accuracy? I thought 1 means the not perturbing the image at all.
2. Overall the the public/private verification part seems to be essential to you contribution and I would recommend emphasizing it better


[1] Müller, A., Lukovnikov, D., Thietke, J., Fischer, A., & Quiring, E. (2025). Black-box forgery attacks on semantic watermarks for diffusion models. In Proceedings of the Computer Vision and Pattern Recognition Conference (pp. 20937-20946).
‏
[2] Jain, A., Kobayashi, Y., Murata, N., Takida, Y., Shibuya, T., Mitsufuji, Y., ... & Togelius, J. (2025). Forging and Removing Latent-Noise Diffusion Watermarks Using a Single Image. arXiv preprint arXiv:2504.20111.‏

**Audience:**

Yes

**Audience Explanation:**

I think the study of  publicly verifiable watermark that is attached to the content is relevant to the TMLR community. Yet, in order for future research to use this work, it is most likely needs to be:
(i) Explained in a clearer way
(ii) Be accompanied with a code for reproducibility purposes.

**Claims And Evidence:**

Yes

**Claims Explanation:**

Most technical claims about robustness are empirically evaluated. Yet, some claims in the intro are too far reaching (see "Additional Comments").

**Requested Changes:**

Please see "Additional Comments" below.

Most importantly, I find that one of the key contributions (the INN) lacks clarity. I'm uncertain both about its usage and about why it represents "the right solution" for this particular application. While I'm willing to accept that it performs well empirically, I would benefit from a clearer explanation of the theoretical or practical motivations behind this architectural choice.

Additionally, I believe several sections of the manuscript would benefit from improved clarity and accuracy of claims.

Finally, regarding the detection aspect, I'm unclear about how the QR code can be successfully inferred when portions of it have been cropped. Specifically, how does the verifier determine which bits are encoded in which locations when part of the code is missing?

---

> ### Author Response · Authors · 2025-12-28
>
> We thank the reviewer for the detailed feedback and would like to address your concerns one by one.
>
> **[The role of INN]**
>
> The INN does not make the watermark robust by learning invariance to perturbations in the way adversarially trained classifiers do. Instead, its contribution lies in enabling exact, bijective encoding and decoding of a high-capacity structured signal, which is critical for cryptographic verification. Robustness in MetaSeal comes primarily from the structured visual encoding (e.g., spatial redundancy and error correction) and frequency-domain embedding with noise-aware training (i.e., adversarial training, mentioned under Eq. 14), while the INN ensures that extraction is stable and information-preserving.
>
> Also, we revised section 5.3 to justify why INN represents "the right solution" for this particular application: its bijective mapping enables exact recovery of large, structured attribution signatures required for public cryptographic verification, and the bijective nature ensures that any semantic manipulation of the image pixels is strictly propagated to the extracted output, providing the inherent tampering sensitivity.
>
> **[Reproducibility]**
>
> We uploaded the code as supplementary materials for the review phase and will release the code on GitHub upon acceptance.
>
> **[QR decoding with cropping]**
>
> QR codes are robust because they contain finder patterns and a built-in error correction mechanism (Reed-Solomon correction). When parts of the code are cropped, if finder patterns remain intact, the decoder can re-localize the grid and recover V.
>
> ---
> 1. IWT stands for Inverse Wavelet Transform. We revised and added clarifications in the caption of Figure 3.
>
> 2.  Although several related works are discussed in Sec. 8, MetaSeal is the first to introduce a distinct and principled solution to the two fundamental questions, i.e., what to embed and how to embed/verify, specifically for reliable image attribution under a forgery-centric threat model. The closest work in terms of attribution objective is Bileve [1], which addresses spoofing-resistant attribution in the text modality; however, its embedding and verification mechanisms are fundamentally different and do not transfer to images. To avoid misunderstanding, we have explicitly added the phrase “under a forgery-centric threat model” in the revision to clarify the scope of our contribution.
>
> 3. We have updated citations in the revision.
>
> 4. We adjust these sentences/sections as follows:
> - The purpose of the text generation is to stabilize semantic granularity. Using a deterministic caption eliminates randomness and ensures the signature is tied to semantic content rather than pixel-level variation. We revised it as `To realize this, we implement $f$ using an image-to-text model that maps visual content to a deterministic textual description, which serves as the high-level semantic summary $M$ and is more invariant to benign image transformations than pixel-level features.`
> - We moved 5.4 to the appendix and only served as high-level analysis.
> - We added clarification: `because it embeds fixed, random watermark bits rather than content-dependent attribution information, and relies on symmetric cryptographic primitives, which require keeping the secret key private. As a result, the verification process cannot be made publicly verifiable without compromising security.`
>
> 5. We revised it as `MetaSeal provides a principled integration of semantic binding with cryptographic verification and exact extraction.`
>
> 6. Figure 3 shows the inference process, so we did not include the loss function, but to improve clarity, we added a reference to the INN loss function into the caption of Figure 3.
>
> 7. To ensure the embedding process remains imperceptible, MetaSeal integrates the Discrete Wavelet Transform (DWT) into the INN architecture. Specifically, the INN takes two inputs: the spatial domain cover image $I$ and the structured spatial secret $V$ (the QR code). Inside the network, $I$ is decomposed into frequency sub-bands. The INN learns a reversible mapping that distributes the information of the spatial QR code $V$ into the frequency features of $I$.  This frequency-domain embedding allows the signal to remain imperceptible in the pixel domain while being robustly preserved. During extraction, the INN inverts this transformation, aggregating the distributed frequency signals back into the coherent spatial structure of the original QR code.
>
>
>
>
> ---
> [1] Bileve: Securing text provenance in large language models against spoofing with bi-level signature. NeurIPS 2024.

---

> > ### Author Response · Authors · 2025-12-28
> >
> > 8. We would like to clarify that these attacks are not directly applicable to MetaSeal due to fundamental differences in the watermarking design. The mentioned attacks target diffusion-based, content-agnostic watermarks that embed a fixed or model-specific signal, which enables estimation, cancellation, or forgery from limited observations. In contrast, MetaSeal embeds content-dependent cryptographic signatures, where each image carries a unique signed message; thus, attacks that rely on reusing a fixed watermark do not directly apply. This further highlights the limitations of prior watermark designs, and our method serves as an urgent fix.
> >
> > The first claim (“Forgery without access to the private key
> > sk is computationally infeasible due to the security of ECDSA”) is a standard cryptographic assumption, referring specifically to the existential unforgeability of the signature scheme under established hardness assumptions. For the second claim, our intent was to reflect an empirical observation rather than a formal security guarantee. We have therefore revised the wording to “mitigating substitute model attacks”, which more accurately captures the scope of our evaluation while leaving room for potential future or yet-unexplored attack strategies.
> >
> >
> > 9. It is not *The closer the scale is to 1 the lower the accuracy*, it shows our method is robust against scaling up but not scaling down, since the latter leads to information loss, which prevents reliable recovery of the embedded secrets and thus degrades accuracy.
> >
> > 10. We revised the contribution part to highlight the support of public verification.

---

> ### Comment · Reviewer_pFdn · 2026-01-04
> **Reviewer Response**
>
> I would like to thank the authors for their response. Below, I detail my remaining concerns, following the numbering above:
>
> 2A. My main remaining issue concerns the novelty claims. The authors claim:
>
> ***“MetaSeal is the first to introduce a distinct and principled solution to the two fundamental questions, i.e., what to embed and how to embed/verify …”***
>
>
> I still find this claim inaccurate. See, for example, the previous works [1,2].
> That said, I do think that combining content-dependent watermarking with public verifiability is novel (to the best of my knowledge), and I would recommend that the authors emphasize this instead.
>
> 2B. In their response to reviewer WbRx, the authors claim:
>
> ***"compared to metadata-based attribution methods, which can achieve anti-forgery but are extremely fragile (e.g., invalidated by format conversion or pixel changes)"***
>
>  Again, this claim seems inaccurate, or at least would require empirical comparison with works such as [1,2].
>
> Additional comments:
>
> 6.  ***”Figure 3 shows the inference process, so we did not include the loss function, but to improve clarity, we added a reference to the INN loss function into the caption of Figure 3.”*** - I would recommend making the figure caption/title clearer in this regard.
>
>
> 7. Thank you for your explanation. It does clarify things for me. I would recommend adding it to the manuscript.
>
>
> 9. Thank you for your explanation. I would recommend explaining it more clearly in the manuscript.
>
>
> [1] Arabi, Kasra, et al. "Seal: Semantic aware image watermarking." Proceedings of the IEEE/CVF International Conference on Computer Vision. 2025.
> ‏
> [2] Evennou, Gautier, et al. "Swift: Semantic watermarking for image forgery thwarting." 2024 IEEE International Workshop on Information Forensics and Security (WIFS). IEEE, 2024.

---

> > ### Author Response · Authors · 2026-01-04
> >
> > Thank you for your follow-up comments and suggestions. We have revised again, and please check the latest version.
> >
> > 2A. Our work shares common ground with [1,2] in that all three employ content-dependent (semantic) watermarking. However, the verification mechanisms differ fundamentally. The methods in [1,2] require access to a secret key at verification time and therefore assume that the verifier is the content owner or a fully trusted party. In contrast, as you noted, MetaSeal supports public verification, which introduces unique design challenges not addressed in [1,2]. **To avoid overclaiming, we have removed the wording “the first” and revised it to “an image attribution watermark”, and emphasized content-dependent watermarking with public verifiability as the key contribution in the introduction.**
> >
> > 2B. We would like to clarify that the statement you quoted refers specifically to metadata-based attribution methods, whose fragility is well recognized in the community (e.g., https://help.openai.com/en/articles/8912793-c2pa-in-chatgpt-images). **The works [1,2] are not metadata-based methods.**
> >
> > We've done modifications based on additional comments:
> >
> > 6. We clarified the figure caption by explicitly stating that Fig. 3 illustrates the inference process of our method.
> >
> > 7. We added the clarification in Section 5.3.
> >
> > 8. We added the explanation in Section 6.5.

---

> > > ### Comment · Reviewer_pFdn · 2026-01-05
> > >
> > > I would like to thank the authors for their quick response. I would also recommend editing the clarifications for the other points (such as point 9). In any case, this is not a concern for me.

---

### Review · Reviewer_WbRx · 2025-12-04

**Summary Of Contributions:**

The paper proposes a method for watermarking that enables attribution of images to their creator or source by combining cryptographic signatures with a content-dependent message. To my knowledge, the overall method is novel:

An image caption and its cryptographic signature are encoded in a QR code. Embedding and detection of watermarks are achieved by an invertible neural network that maps an image together with its caption and cryptographic signature in QR code format to a watermarked image (plus a noise term that encodes the lost information).

The proposed method thereby addresses robustness to forgery in several ways. Firstly, tampered images can be detected by use of the invertible neural network, as tampering leads to a degradation of the QR codes. Secondly, the matching key can decode the cryptographic signature to authenticate the message. Since the message depends on the content, the reuse of watermarks on new (possibly harmful) data can be detected. By combining these capabilities, the approach is interesting in certain use cases.

&nbsp;

**Strengths:**
+ The content dependence of the watermark ensures that attempts to reuse the watermark on unrelated images can be detected.
+ The model is shown to be robust to forgery. An interesting experiment illustrates how tampering of watermarked images becomes visually apparent in form of corrupted QR codes.
+ The watermarking is a post-hoc method, making it applicable to both natural and AI-generated content (rather than being restricted to generated content).
+ The experimental section demonstrates strong attribution performance for long message lengths.

&nbsp;

**Weaknesses:**
- The method is not placed well into related work in several ways: (i) It does not acknowledge existing work on watermarking that similarly use invertible neural networks. (ii) Limitations of prior work discuss only those of a selectively chosen list of baselines. (iii) The experimental section has a selected list of baselines
- The robustness of the method is very limited
- The theoretical analysis does not provide meaningful theoretical insights into the proposed method

&nbsp;

**Discussion of perceived weaknesses:**

*Related work:*

- The proposed solution is not positioned within the broader literature on fingerprinting, which similarly allow to attribute content to specific users or models. Most importantly, although the paper highlights invertible neural networks as a key choice for achieving strong performance with large capacity, it does acknowledge prior work that uses invertible networks for watermarking, e.g. [1-3]. In particular, [3] also use invertible neural networks for embedding large capacity messages into images, and it builds upon earlier “high-capacity” solutions that hide entire images inside other ones.
- Similarly, the Section 3.2 (“Limitations of previous work”) compares to “Hidden” a method suggested in 2018 and, to my knowledge, the first paper on watermarking to jointly train an encoder and a decoder. There exist more recent methods with larger capacity, e.g. [2-4], so the stated limitations of previous work seem to apply only to a selectively chosen list of baselines.
- The experimental section also compares against a restricted list of models, most of which are older approaches. There is an emphasis on the high-capacity of the proposed method. However, the only recent baseline, WAM, is designed for retrieval of the watermark from any segment of the image, and it therefore does not target high-capacity watermarking.

[1] Ma et al, Towards Blind Watermarking: Combining Invertible and Non-invertible Mechanisms (ACM Multimedia 2022)

[2] Jing et al, Hinet: Deep image hiding by invertible network (ICCV 2021)

[3] Lu et al, Large-capacity image steganography based on invertible neural networks. (CVPR 2021)

[4] Yu et al, Responsible disclosure of generative models using scalable fingerprinting (ICLR 2022)

&nbsp;

*Robustness:*

+ The experiments show limited robustness to natural perturbations. For example, even small payloads fail under jpeg compression at 80% compression and additive noise of only 0.06 is sufficient to break the watermark. By construction, the proposed method is also likely not robust under cropping, but this is not investigated.
Due to its use of a neural network, the method may also lack robustness to adversarial examples. Finally, because the watermark only adds minimal distortions to an existing image, the approach is not robust under noise-denoising strategies

&nbsp;

*Theoretical analysis:*

+ Section 5.4 promises to offer a theoretical analysis of the method and its design choices. However, the arguments in Part 1 seem unclear and appear to refer to the use of QR codes, which are not introduced until the experimental section. It is also no explained whether Equation (17) follows from a concrete derivation or is presented merely as an intuition.  Part 2 discusses the inherent error correction of QR codes (again, which are not mentioned as part of the methodology), but these properties are not specific to the proposed method: Fingerprints can always incorporate error correction mechanisms.

**Audience:**

Yes

**Audience Explanation:**

The topic of watermarking and detection of forgery are very active research topics in the machine learning community and there is a great interest in new methodologies.

**Broader Impact Concerns:**

No concerns

**Claims And Evidence:**

No

**Claims Explanation:**

While most claims are well-supported, there are the following issues related to the above mentioned weaknesses.

**Positioning of paper and contribution:**

Given the above, it remains unclear why the paper claims being “*the first image attribution watermark that provides innovative solutions to these fundamental questions*”. The discussion of limitations of related work and experimental section do not seem to survey a fair acknowledgement and comparison to existing work.  The method could be better positioned with a more complete review of related work and a fairer and a more modest discussion of the advantages.

**Robust against benign transformation**

The method is advertised as being robust, in particular against benign transformations. The experiments do not support this claim (see discussion of weaknesses).

In addition, the paper argues that other methods that rely on detector-based verification can be tricked. However, the proposed method also relies on an (invertible) neural network and might therefore also be vulnerable to adversarial examples. Elsewhere, the paper suggests that robustness to adversarial removal of the watermark would imply “*tolerance to modifications, which contradicts the goal of attribution*” and that removal of watermarks poses minimal harm. To the latter, I disagree:  Adversarial attacks (PGD) can preserve image quality and content while removing attribution, enabling the removal of watermarks from harmful content or to evade copyright-related attribution.

**Requested Changes:**

**Major:**

Based on my discussion above, I suggest the following changes as critical:

+ The paper should acknowledge similar approaches in prior work, e.g. by incorporating (at least some of) the above-named papers on watermarking using invertible networks and large capacity watermarking techniques. The paper should work out the contributions relative to existing work.
+ The paper should tone down its claims on robustness.
+ The method should be fairly classified in the existing literature in terms of robustness and capacity.
+ The paper should either introduce the use of QR codes in the methodology or formalize what generalization the author’s have in mind and clarify Part 1 in the theoretical analysis.

**Minor:**

+ Definition 2 lacks an explanation of its notation
+ In Tables 2 and 3, the best entry in each column should be in bold, not just the best performance of the proposed method

---

> ### Author Response · Authors · 2025-12-27
>
> Thank you for your time and feedback. Below we describe the revisions and clarifications made in response to the review.
>
> **[Existing works and positioning of our work]**
>
> The “missing” related works mentioned in the review comments fall into three related but **distinct** areas: steganography [2,3], fingerprinting [4], and watermarking [1]. *We believe there could be some confusion among these fields and would like to highlight their distinctions.* **Steganography** aims at covert communication, prioritizing large capacity and resistance to steganalysis, but it is typically fragile (even to benign transformations) and does not support reliable attribution. **Watermarking**, in contrast, assumes the presence of an embedded signal and seeks to bind information to content; however, existing methods, including INN-based ones such as [1], generally embed small, content-agnostic, and semantically meaningless bit strings, making them vulnerable to replay and forgery. **Fingerprinting** achieves attribution by modifying model weights so that generated images exhibit identifiable characteristics [4], whereas watermarking embeds secrets directly into images and does not rely on model-specific behavior; both typically support only limited payloads (e.g., 128 bits in [4] and 30 bits in [1]).
>
> MetaSeal is distinct from all three lines of work. **Our contribution is not to propose a large-capacity watermark per se, but to identify what should be embedded to achieve anti-forgery attribution, i.e., content-dependent cryptographic signatures rather than random bits, and how to embed them reliably.** Large capacity is a requirement imposed by anti-forgery attribution, not the goal itself. Invertible neural networks serve as an enabling tool to support exact extraction under this requirement. **We already cited HiNet [2] in the original manuscript and have incorporated additional INN-related works in the revised Section 5.3 and discussion on steganography and fingerprinting in related work (Section 8) to better reflect the positioning of our contribution.**
>
> **[Robustness]**
>
> For cropping, we in fact investigated it and showed the results in the appendix (original Figure 11, now Figure 12 in revision). Our method maintains robustness under cropping to a certain extent due to the built-in error correction mechanism in QR code.
>
> As the reviewer jAXW correctly noted, there is an inherent trade-off between robustness and unforgeability: increasing tolerance to transformations can enlarge the set of modified images that still verify, potentially raising the risk of misattribution. Our design therefore prioritizes unforgeability over aggressive robustness. **To address this concern, we have toned down the claims on robustness in the contribution section and added an explicit discussion in the Limitations section.** More broadly, this reflects an open challenge for future work: improving robustness to benign transformations inevitably increases overall tolerance, which may expand the set of modified images that still verify and thus raise the risk of misattribution. How to achieve selective robustness, i.e., strong robustness to certain benign transformations while remaining sensitive to others, remains an important and unresolved problem in attribution watermarking.
>
> **[MetaSeal vs detector-based verification]**
>
> We clarify that although MetaSeal uses an INN as part of the embedding/extraction pipeline, **its security does not rely on a neural detector decision, which is the core vulnerability exploited by adversarial examples in prior work.** In detector-based schemes, adversarial perturbations (e.g., PGD) can directly induce false positives, causing an unwatermarked or unrelated image to be classified as authentic. In contrast, MetaSeal uses the INN only as a reconstruction mechanism; attribution validity is ultimately determined by cryptographic verification of the recovered signature, which adversarial perturbations cannot fabricate or reconstruct.
>
> **[Other revision]**
>
> 1. We introduced the use of QR codes in the methodology.
>
> 2. We refined Definition 2 with a connection to our watermarking settings and added more explanations.
>
> 3. We refined Tables 2 and 3 as suggested.
>
> ---
> We sincerely appreciate your suggestions that help us improve our work, and hope the updated revision has addressed your concerns.

---

> > ### Comment · Reviewer_WbRx · 2025-12-31
> > **Follow up to author's response**
> >
> > I thank the authors for their clarifying response and their adjustments in the paper reflecting some of my comments.
> >
> > My concern on the related work remains, as the choice of baselines (given that MetaSeal is distinct from methods used in watermarking, fingerprinting and steganography) remains unclear to me. Possibly the authors could clarify why the selected baselines are representative/SOTA for the task?
> >
> > On my concern about claims on robustness: The paper emphasizes how attribution watermarking requires sufficient robustness to benign transformations used in real-world applications. The paper also presents the fragility to benign transformations as a weakness of related work.  Could the authors please clarify how the experimental results show that their proposed solution is sufficiently robust to benign real-word distortions?
> >
> > Minor points:
> > - The theoretical analysis was moved to the appendix, but without improvements to its content.
> > - Definition 2 still contains undefined notation. I would also suggest to make the definition self-contained.

---

> ### Author Response · Authors · 2025-12-31
>
> Thank you for the follow-up comments.
>
>
> **[Baseline Justification]**
>
> - General Deep Watermarking (HiDDeN, RivaGAN): These are the standard benchmarks for learning-based watermarking robustness and imperceptibility. We included them to demonstrate a critical limitation: while effective at low capacities (e.g., 30 bits), they fail to converge when trained for the high capacities (512 bits) required for cryptographic signatures (as shown in Fig. 2).
>
> - Generative Watermarking (Stable Signature): This is representative work for watermarking AIGC directly during generation. We compare against it to highlight MetaSeal's advantage in being model-agnostic (post-hoc) and supporting significantly larger payloads than Stable Signature's fixed latent embedding.
>
> - Tamper Localization (WAM): This is the most recent SOTA method for localizing edits. We compare against WAM to demonstrate that MetaSeal can achieve comparable tamper evidence features while simultaneously carrying a payload that is orders of magnitude larger (88x).
>
> - Industrial Standard (DwtDctSvd): This is the legacy method currently deployed in Stable Diffusion. We compare against it to show that MetaSeal offers superior payload and recovery accuracy compared to the method currently used in production systems.
>
> These baselines span the dominant design paradigms in current image watermarking: classical signal-processing methods, neural fixed-bit watermarking, adversarially trained robust watermarking, in-generation watermarking, and localization watermarking. **By comparing against them, we demonstrate that existing methods either fail to scale to cryptographic payload sizes, suffer from degraded extraction accuracy, or rely on assumptions incompatible with forgery-resistant attribution (fixed, content-agnostic watermark with detector-based verification).**
>
> More importantly, anti-forgery watermarking for reliable image attribution remains understudied. Most recent works continue to follow the same design principles as the above baselines, while remaining vulnerable to forgery attacks or being difficult to adapt to cryptographic verification. **Our results show that achieving both large capacity and near-perfect extraction accuracy cannot be obtained by incremental improvements to existing methods, which motivates the need for an entirely new framework such as MetaSeal.**
>
> If you have specific methods in mind that you believe could challenge our conclusions, please let us know, and we will include them as well.
>
> **[Robustness]**
>
> In our setting, robustness is defined relative to forgery-resistant attribution, rather than traditional removal-resistant watermarking. In particular, compared to metadata-based attribution methods, which can achieve anti-forgery but are extremely fragile (e.g., invalidated by format conversion or pixel changes), MetaSeal provides improved robustness while preserving cryptographic binding. For the most common real-world distortion, JPEG compression, standard photo uploads to social media (X, LinkedIn, etc.) typically use quality levels around 85, while MetaSeal maintains perfect verification accuracy for JPEG quality>84. In contrast to steganographic methods, which are typically fully fragile to benign distortions, MetaSeal is better characterized as a semi-fragile watermark, balancing robustness and sensitivity in support of reliable attribution.
>
> Being sufficiently robust to benign transformations is ideal, and our method has shown potential to be robust to a certain extent. We acknowledge that for a commercial-grade real-world application, robustness could be pushed further via more extensive adversarial training. However, **as a research prototype, MetaSeal successfully demonstrates that it is possible to decouple "cryptographic security" from "metadata fragility," offering a new direction for robust attribution.**
>
> **[Analysis Improvement]**
>
> We revised this section to clarify its scope and intent. The section is now explicitly framed as an intuitive and probabilistic design analysis, rather than a formal security proof. We introduce the use of QR codes in the methodology, and explain why this approach is fundamentally different from simply adding error-correcting codes to bit-wise fingerprints. The revised analysis makes clear how visual structure, spatial redundancy, and global decoding jointly enable reliable recovery of large cryptographic payloads. Please check the updated version.
>
> **[Notation Definitaion]**
>
> Upon review, the only two notations we found that are commonly used but not explicitly defined are
> Pr[⋅], which denotes probability, and
> negl(λ), which denotes a negligible function in the security parameter
> λ. Are these the notations you were referring to, or are there others that we should clarify to make the definition fully self-contained?

---

> > ### Comment · Reviewer_WbRx · 2026-01-05
> >
> > I thank the authors for their reply. I acknowledge that the major revision points raised in my previous review have been partially addressed.
> >
> > **Robustness:** I thank the authors for the clarifications provided. Nevertheless, the experimental results continue to show a marked performance degradation under relatively modest noise perturbations. This sensitivity should be acknowledged as a limitation, as it weakens the robustness claim in its current form.
> >
> > In Figure 7, I suggest replacing the term “JPEG intensity” with “JPEG quality,” as the former is not standard terminology.
> >
> > **Baselines:** The paper compares against Ma et al., arguing that while their method leverages invertibility for improved extraction or tamper localization, it embeds fixed, content-agnostic bit strings and is therefore vulnerable to replay and cross-image forgery. However, it is unclear that this limitation is inherent to that approach, as the watermark could in principle be made content-dependent.
> >
> > Both the discussion of limtations of prior work and the experimental evaluation emphasize limited capacity as the main shortcoming of the baselines. In this context, it seems appropriate to consider methods that already achieve substantially larger capacity, such as those of Lu et al. and Ma et al.
> >
> > The current experiments, as well as the theoretical analysis, do not provide evidence supporting the claim that achieving both large capacity and near-perfect extraction accuracy cannot be obtained through incremental improvements to existing methods. If such an impossibility result or formal justification exists, I appreciate a clarification which could strengthen the paper.
> >
> > **Formal definition:** Without intending to argue this point in detail, I offer the following clarification regarding Definition 2: The definition relies on several elements that are not formally defined and are presumed to be familiar to the reader (items (i)–(v) below). Since the subsequent analysis does not rely on the specifics of ECDSA, it is not clear what is gained by including a partially informal definition rather than treating ECDSA as a fully established component.
> >
> > (i) "probabilistic polynomial-time adversary" is not formally defined (ii) negl(lambda) is not defined to denote a negligible function (which itself is not defined) (iii) The sentence on the set Q "is initialized to empty set, records all semantic feature messages queried by A to the signing oracle" is not a formal statement (iv) The initialization of Q appears after checking for a message M in Q. (v) Notation A^{O_Sign} is not defined

---

> ### Author Response · Authors · 2026-01-07
>
> Thank you for continuously helping us improve our work. We've done another round of revision, and please check the latest version.
>
> **Robustness**: We have acknowledged it in the limitations section.
>
> We replaced the term “JPEG intensity” with “JPEG quality”.
>
> **Baselines**: Ma et al. do not support large payload capacity (64 bits as reported); their use of INNs primarily targets high imperceptibility. Besides, Lu et al. and Jing et al. (HiNet) demonstrate that INNs can support very large payloads with high fidelity, but they do so in a steganographic setting with essentially ZEOR robustness. While these methods could potentially be extended with adversarial or robustness-oriented training, doing so would require embedding stronger signals, which would inevitably degrade imperceptibility, highlighting the **inherent tension between robustness and invisibility at high payloads.** In fact, our INN is built upon HiNet (as described in Appendix B). The reason our robustness is intentionally limited is that we explicitly consider the trade-off between robustness and imperceptibility, and choose to sacrifice robustness beyond mild transformations in order to preserve the visual quality of the watermarked image.
>
> Please note,  our contribution is not the invention of individual modules, but the end-to-end design that integrates them in a new way to address an urgent issue: we propose the Visual Attribution Signature (content-dependent, cryptographically meaningful information carried as a structured visual pattern QR code) and identify INNs as an effective embedding–extraction mechanism for anti-forgery image watermark. As with building using existing Lego blocks, the novelty lies in how these components are assembled to address the under-studied problem of image watermark forgery, rather than in claiming the blocks themselves are new.
>
> **Formal definition:** We originally included Definition 2 to explain the source of unforgeability in our method. However, following the reviewer’s suggestion, we have removed this definition and instead treat ECDSA as a fully established cryptographic primitive, citing its standard security guarantees (e.g., [1]).
>
>
> ---
> [1] Goldwasser, Shafi, Silvio Micali, and Ronald L. Rivest. "A digital signature scheme secure against adaptive chosen-message attacks." SIAM Journal on computing 17.2 (1988): 281-308.

---

> > ### Comment · Reviewer_WbRx · 2026-01-07
> >
> > I thank the authors for their thoughtful clarifications during the discussion and their efforts to revise the manuscript in response to my comments and concerns.

---

### Review · Reviewer_jAXW · 2025-12-18

**Summary Of Contributions:**

The paper proposes MetaSeal, a content-dependent watermarking framework for reliable image attribution that targets two key weaknesses of prior approaches: content-agnostic watermarks and detector-based verification that can be attacked. MetaSeal derives semantic features from the image, digitally signs them, encodes the signature into a QR code, and embeds this pattern into the image using an invertible neural network so the secret can be recovered precisely and verified cryptographically with a public key. Experiments on natural and AI-generated images show perfect extraction/verification at much larger payloads than common baselines, robustness to benign transformations (e.g., JPEG), sensitivity to malicious edits with visible tamper evidence, and strong resistance to replay, mixup, PGD-style, and adaptive forgery attempts.

**Audience:**

Yes

**Audience Explanation:**

Researchers working on trustworthy AI, digital watermarking, and content provenance would be interested in this paper’s findings, as it addresses forgery-resistant image attribution with cryptographic guarantees and empirical evaluation relevant to both natural and AI-generated images.

**Claims And Evidence:**

Yes

**Claims Explanation:**

Strengths include a principled cryptographic design that tightly binds attribution to image semantics, enabling high-capacity yet exact watermark recovery, along with a thorough empirical evaluation that probes multiple dimensions (image quality, robustness to benign transformations, tamper sensitivity, and resistance to several forgery/adaptive attack settings).

**Requested Changes:**

1. Some of the theoretical presentation feels like "decorative math.” The paper largely restates the standard definition of cryptographic unforgeability without clearly connecting it to the watermarking setting. Similarly, the analysis in Sec. 5.4 is fairly superficial and does not provide much additional insight beyond a high-level comparison. I’m completely fine with omitting it or moving it to the appendix.


2. The paper’s discussion of robustness versus forgery resistance in Sec. 2 is not entirely accurate. As currently written, it seems to imply that attribution watermarking does not require robustness to strong removal attacks, which may not hold under realistic threat models. Indeed, there is an inherent trade-off between robustness and unforgeability: increasing tolerance to transformations can enlarge the set of modified images that still verify, potentially raising the risk of misattribution. It may be better to frame the distinction as requiring a different balance (and threat-model-dependent prioritization) rather than suggesting robustness is largely unnecessary.


3. It is unclear how the method performs under regeneration-style watermark removal attacks (see references below). Some similar plots as in Sec. 6.5.2 would be enough to help us understand the robustness of the proposed approach in this new challenge.

Zhao, Xuandong, Kexun Zhang, Zihao Su, Saastha Vasan, Ilya Grishchenko, Christopher Kruegel, Giovanni Vigna, Yu-Xiang Wang, and Lei Li. "Invisible image watermarks are provably removable using generative AI." Advances in neural information processing systems 37 (2024): 8643-8672.

Liu, Yepeng, Yiren Song, Hai Ci, Yu Zhang, Haofan Wang, Mike Zheng Shou, and Yuheng Bu. "Image watermarks are removable using controllable regeneration from clean noise." arXiv preprint arXiv:2410.05470 (2024).



Minor point:
In Eq. (5), the indicator function is confusingly written. I believe the intended meaning is to count an error when the predicted bit crosses the 0.5 threshold but disagrees with the ground-truth bit (i.e., $\mathbf{1}[\mathbf{1}(p(\hat m_i)\ge 0.5)\neq m_i]$). The current form does not express this clearly/correctly and should be rewritten.

---

> ### Author Response · Authors · 2025-12-27
>
> Thanks for your constructive feedback! We have made the following changes to address your concerns.
>
> 1. We moved the theoretical analysis to the appendix and renamed it “Analysis of Structured Visual Embedding” to provide a high-level analysis of the benefits of structured visual embedding. We also revise definition 2 to connect cryptographic unforgeability with the watermark setting.
>
> 2. We revised section 2.2 and highlighted that the attribution watermark requires a different trade-off between anti-forgery and robustness.
>
> 3. We added the performance of the regeneration attack in Appendix C, where the results show that the regeneration attack can make the recovered QR code unscannable, at a great cost of image quality (measured by PSNR and SSIM).
>
> 4. We corrected the indicator function in Eq. (5).

---

### Author Response · Authors · 2025-12-28
**Revision Summary**

We thank all reviewers for their constructive feedback. In response, we have revised the paper to improve clarity, positioning, and accuracy of claims. **The revised version of the paper has been uploaded. We have also included, as the second part of the uploaded file, a version with all modifications highlighted. Reviewers may refer to the clean version for the final presentation and to the highlighted version for quickly locating the changes.**

The main changes are summarized as follows:

1. **Clarified positioning and related work.**
   We revised the related work section to clearly distinguish image attribution watermarking from steganography and fingerprinting, and to acknowledge prior INN-based and high-capacity hiding methods. We clarified that our contribution is not large-capacity watermarking per se, but a forgery-centric attribution framework that identifies *what to embed* (content-dependent cryptographic signatures) and *how to embed/verify* them with public verifiability.

2. **Refined contribution and novelty claims.**
   We clarified that MetaSeal leverages existing techniques in a principled integration rather than introducing each component in isolation. We explicitly scoped our contribution to reliable image attribution under a forgery-centric threat model, and highlighted that MetaSeal enables publicly verifiable attribution using asymmetric cryptography.

3. **Toned down and clarified robustness claims.**
   We limited robustness claims to benign, semantics-preserving transformations to a certain extent and added a discussion of the inherent trade-off between robustness and unforgeability. We clarified that MetaSeal prioritizes forgery resistance over aggressive robustness and that selective robustness remains an open challenge.

4. **Clarified the role of the INN and DWT/IWT.**
   We revised the caption of Fig. 3 and section 5.3 to explain the function of the INN, its training objective, and how DWT/IWT are used to embed structured secrets in the frequency domain. We also provided justification for why INN is a suitable choice in section 5.3.

5. **Removed theoretical analysis.**
  We moved the theoretical analysis to the appendix and renamed it “Analysis of Structured Visual Embedding” to provide a high-level analysis of the benefits of structured visual embedding.

6. **Refined Definition 2.**
   We removed Definition 2 and treat ECDSA as a fully established cryptographic primitive, citing its standard security guarantees.

Overall, these revisions significantly improve the clarity, accuracy, and positioning of the paper, and we believe they address the reviewers’ concerns while strengthening the presentation of our contributions.

---

### Decision · Action_Editor_QgtJ · 2026-01-20

**Recommendation:** Accept as is

**Audience:**

Yes

**Audience Explanation:**

All reviewers agree that the submission would be interesting to TMLR's audience.

**Claims And Evidence:**

Yes

**Claims Explanation:**

All reviewers agree that the submission is supported by sufficient evidence.